# Sorting at embryonic boundaries requires high heterotypic interfacial tension

Laura Canty[1], Eleyine Zarour[1], Leily Kashkooli[1,2], Paul François[1,3] & François Fagotto[1,2,4]

The establishment of sharp boundaries is essential for segregation of embryonic tissues during development, but the underlying mechanism of cell sorting has remained unclear. Opposing hypotheses have been proposed, either based on global tissue adhesive or contractile properties or on local signalling through cell contact cues. Here we use ectoderm–mesoderm separation in *Xenopus* to directly evaluate the role of these various parameters. We find that ephrin-Eph-based repulsion is very effective at inducing and maintaining separation, whereas differences in adhesion or contractility have surprisingly little impact. Computer simulations support and generalise our experimental results, showing that a high heterotypic interfacial tension between tissues is key to their segregation. We propose a unifying model, in which conditions of sorting previously considered as driven by differential adhesion/tension should be viewed as suboptimal cases of heterotypic interfacial tension.

[1] Dept. of Biology, McGill University, Montreal, QC, Canada H3A1B1. [2] CRBM, CNRS, Montpellier 34293, France. [3] Dept. of Physics, McGill University, Montreal, QC, Canada H3A2T8. [4] Dept. of Biology, University of Montpellier, Montpellier 34095, France. Correspondence and requests for materials should be addressed to F.F. (email: francois.fagotto@crbm.cnrs.fr)

Physical separation of embryonic cell populations is fundamental to metazoan development. The process, which results in the sharp delimitation of cell masses by so-called tissue boundaries, appears to rely on the ability of individual cells to distinguish between homotypic contacts, i.e. contacts with cells of the same type, and heterotypic contacts with cells of a different type. This property can be shown by mixing dissociated cells from different embryonic regions and observing their progressive sorting into separate groups. These observations led Holtfreter to propose the concept of 'selective cell affinities'[1, 2]. Four major models have attempted to explain the elusive nature of these affinities: the differential adhesion hypothesis (DAH)[3] stated that different cell types would sort according to their respective intercellular adhesive strength to maximise the number of adhesive complexes formed. In the differential interfacial tension hypothesis (DITH), Brodland[4, 5] introduced contractility of the cortical actomyosin cytoskeleton as an essential parameter of cell–cell interactions. Tenants of the DITH have argued that tissue differences in cortical tension are the major driver of tissue separation[6, 7]. The selective adhesion hypothesis (SAH) proposes that tissue segregation is due to the expression of unique sets of cadherins, which are considered to bind homophilically[8, 9]. Lastly, cell surface cues, such as ephrin ligands and Eph receptors have been involved in the generation of repulsive reactions at heterotypic contacts (reviewed in ref. [10]). At the cellular level, these reactions are characterised by a local increase in cortical actomyosin contractility, and consequently destabilisation/disruption of cell adhesion at heterotypic contacts[11].

These four models can be expressed and directly compared using the concept of interfacial tension[4, 5] (Fig. 1a). Note that to avoid ambiguities, we prefer to use the term 'contact tension' and reserve the term 'interfacial tension' to the tension at tissue interfaces[12, 13] (Fig. 1b). DAH and DITH can be expressed by a similar configuration, where the tension at homotypic contacts is higher in one of the two cell populations and intermediate at heterotypic contacts (Fig. 1c). Ephrin-Eph-mediated repulsion creates a different situation, where tension is strongly increased at heterotypic contacts compared to homotypic contacts inside the tissues (Fig. 1c). We call this configuration 'high heterotypic interfacial tension' (HIT). Most experimental data support ephrin-Eph-dependent HIT as the major mechanism for separation in vertebrates[10, 14–22], and evidence for HIT has also been found in Drosophila[23–25], indicating that it may be a general feature of tissue separation. Note that mechanisms other than ephrin-Eph-mediated repulsion can also produce HIT (see 'Discussion'). In particular, SAH can be viewed as a case of HIT (Fig. 1c).

However, embryonic tissues also differ in adhesive and tensile properties[6], leaving open the possibility of a contribution of DAH/DITH to the process. Evidence supporting a role for these differences has remained so far largely correlative. In fact, ectoderm–mesoderm separation in Xenopus has turned out surprisingly resistant to manipulations of cadherin levels[26, 27]. Alternatively, adhesive and tensile properties may be participating in separation by reinforcing repulsion-based separation. Cell–cell adhesion and ephrin-Eph signalling are indeed involved in an intimate interplay: we have shown that a proper balance between ephrin-Eph signalling, and adhesion is crucial in setting the threshold required for overt cell detachments, both at the tissue boundary interface and within the tissues[20, 21]. Thus, the impact of adhesive and contractile differences on tissue separation remains unclear.

In Xenopus, separation of mesoderm from ectoderm is controlled by a well-characterised network of ephrins and Eph receptors[20, 21]: both tissues express multiple ephrins and Eph receptors, but repulsion is restricted to the ectoderm–mesoderm contacts, due to the asymmetric expression of selected ephrin-Eph pairs that trigger bidirectional repulsive signals. The

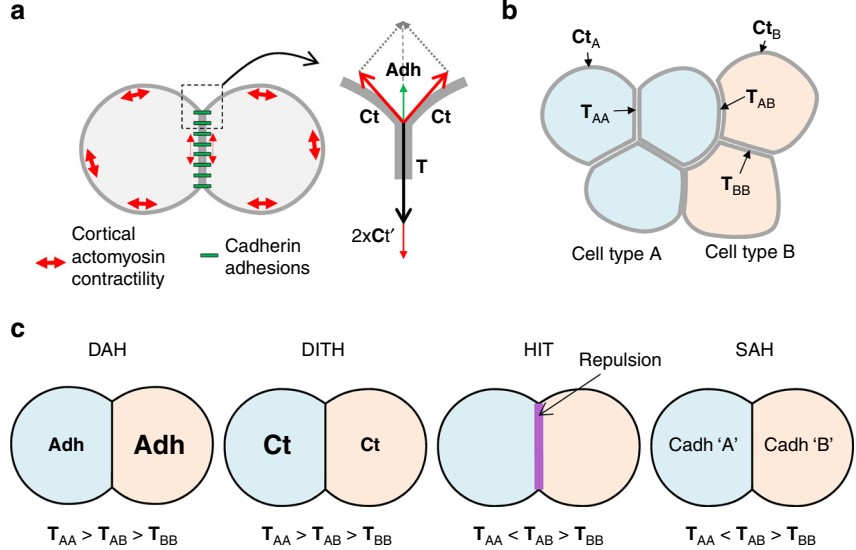

**Fig. 1** Contact tension and cell sorting. **a** Diagram of two adhering cells and representation of the force equilibrium at a contact vertex. The cadherin adhesive structures are represented in *green,* and the contractility of the actomyosin cell cortex by *red double arrows.* Note that cadherin adhesions influence the cell cortex, decreasing tension along contacts (*smaller double arrows*). The equilibrium of forces at a vertex involves the cortical tension **Ct** at the free surface of each cell and the contact tension **T** between the two cells. **T** is the sum of the two cortical tensions at the contact (**Ct′**) and of cell–cell adhesion (**Adh**), which acts in the opposite direction to expand the contact. **b** Contact tensions in tissues. $T_{AA}$ and $T_{BB}$ represent tensions at homotypic contacts. $T_{AB}$ represents the contact tension at heterotypic contacts, also called here interfacial tension. **c** Comparison of the four models for cell sorting and separation based on contact tensions. In the differential adhesion hypothesis (*DAH*) and the differential interfacial tension hypothesis (*DITH*), the two cell populations have different homotypic tensions, and the heterotypic tension is intermediate. Repulsive mechanisms such as those generated by ephrin-Eph signalling generate high interfacial tension (*HIT*). The same situation can be achieved in the selective adhesion hypothesis (*SAH*), due to preferential homotypic cadherin interactions

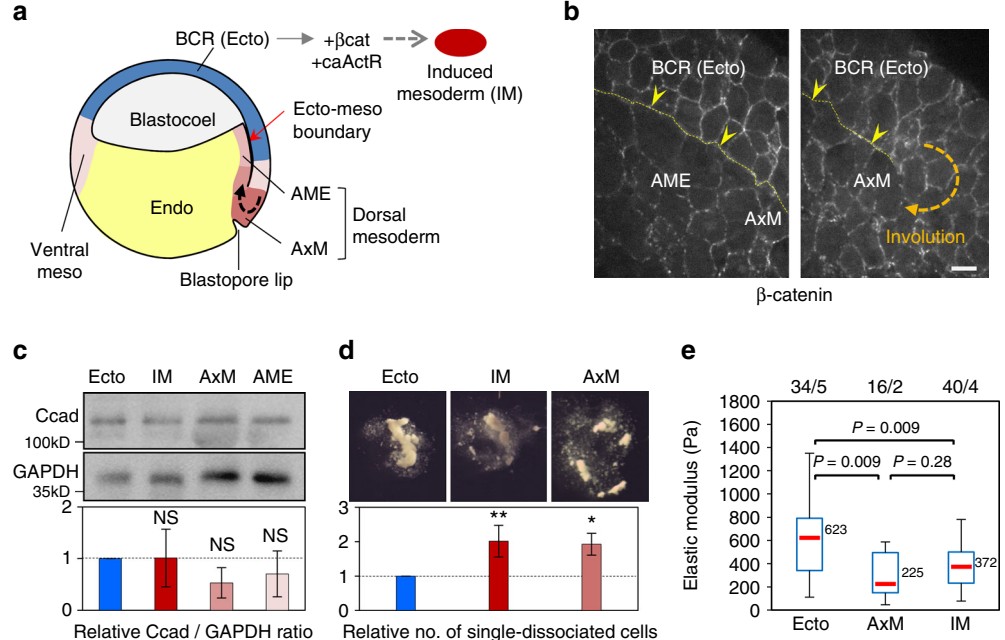

**Fig. 2** *Xenopus* ectoderm and mesoderm and their adhesive and contractile properties. **a** Diagram of the early *Xenopus* gastrula indicating the regions used as the source for tissue explants. Induced mesoderm was produced by expression of β-catenin (β-cat) and constitutively active Activin receptor (caActR) in the blastocoel roof (*BCR*). **b** Distribution of adhesive structures in the ectoderm blastocoel roof (*BCR* or *Ecto*), anterior mesendoderm (*AME*) and axial mesoderm (*AxM*) visualised by β-catenin immunostaining. The thin *yellow dotted line* and *yellow arrowheads* point to the mature (*left panel*) or nascent (*right panel*) ectoderm–mesoderm boundary. Scale bar, 20 μm. See also Supplementary Fig. 1b (**c**–**e**). Characterisation of adhesive and contractile properties. **c** Total cadherin levels in dissected tissues determined by immunoblot and expressed as the relative ratio to GAPDH. Graph shows mean values from seven independent experiments. Error bars, s.d. NS: not significantly different from ectoderm, based on one-sided Student's *t*-test. **d** Resistance to dissociation, measured as the number of single-dissociated cells, normalised as the ratio to ectoderm. Mean of four independent experiments. Individual comparisons were made using one-sided Student's *t*-test. *$P < 0.05$; **$P < 0.01$. Error bars, s.d. **e** Cortical tension. The elastic modulus of single cells was determined using AFM. The box plots show the interquartile range (*box limits*), median (*centre line* and corresponding value), and min and max values without outliers (*whiskers*). Number of cells/experiments are indicated on top. Individual comparisons were done using Tukey's HSD test after a significant one-way ANOVA ($P = 5.7e{-}08$)

complementarity between ephrinB3 and EphA4, expressed in the ectoderm and in the mesoderm, respectively, is particularly important in establishing strong repulsions between these tissues[21]. Another important pair is formed by mesoderm-enriched ephrinB2 and ectoderm-enriched EphB4[21]. Other components, such as ephrinB1, are more widespread but also contribute significantly to build up sufficiently high repulsion at the tissue interface to overcome cadherin-mediated adhesion[20, 21]. Repulsive signals also occur within the mesoderm, but are not sufficiently strong to cause cell detachment, unless cadherin levels are artificially lowered[21]. Ephrin-Eph signalling in the ectoderm is weak and has a proadhesive activity[20]. The two tissues have also distinct physical characteristics: cohesion (the global tissue property reflecting the strength of cell–cell adhesion) tends to be higher for ectoderm than mesoderm[28, 29]. Furthermore, single ectoderm cells characteristically bleb, whereas mesoderm cells do not[30], suggesting that the former have higher cortical tension.

In this study, we ask two important questions: are differences in adhesive strength or cortical contractility necessary and/or sufficient for tissue separation? Could these differences cooperate with ephrin-Eph-mediated repulsion, in other words, could a combination of DAH/DITH and HIT be more effective than HIT alone? To address these questions, we directly compare the impact of ephrin-mediated repulsion and of differences in adhesion/contractility in a series of assays that test maintenance of ectoderm–mesoderm separation, induction of separation between tissues of the same origin and cell sorting from two mixed cell populations. We find that HIT fully accounts for cell sorting and tissue separation in this system, with little to no contribution from adhesive/tensile differences. Computer simulation supports the general conclusion that establishment of a sharp tensile discontinuity at the tissue interface is the key parameter for tissue segregation.

## Results

### Ectoderm and mesoderm adhesive and contractile properties.

Throughout this study, we used ectoderm and mesoderm from the early gastrula as well as induced mesoderm (IM) produced by ectopic activation of the Wnt and TGFβ signalling pathways in the ectoderm (Fig. 2a). RT-PCR confirmed that this induced tissue expressed a mixture of markers for various sub-regions of dorsal mesoderm (Supplementary Fig. 1a). As a prerequisite to this study, we compared the adhesive and contractile properties of these tissues. Total C-cadherin levels were similar in ectoderm and IM but tended to be lower in endogenous mesoderm, although differences varied greatly between embryo batches (Fig. 2c). Cell surface adhesion complexes were visualised by β-catenin and C-cadherin immunolabelling (Fig. 2b; Supplementary Fig. 1b). Both signals were high in the inner ectodermal layer and at the blastopore lip but progressively decreased in the mesoderm toward the anterior. To compare adhesion in different tissues, we used a dissociation assay (Fig. 2d). Endogenous mesoderm and IM appeared to be less resistant to dissociation than the inner ectoderm cells, consistent with previous measurements of global tissue cohesion[12, 28] and of cell–cell reaggregation[31]. Actomyosin contractility confers rigidity to the cell cortex, which can be

measured on single cells by atomic force microscopy (AFM). We found that ectoderm cells had a higher elastic modulus than mesoderm and IM cells (Fig. 2e). In summary, ectoderm cells were found to be both stiffer and more adherent than mesoderm cells, and IM was similar to endogenous mesoderm for both parameters, validating its use in our subsequent experiments.

**Manipulation of adhesiveness and actomyosin contractility.** We established conditions to manipulate cell–cell adhesion and

cortical stiffness, with the goal to either level the differences observed between ectoderm and mesoderm, or, on the contrary, re-create these differences. The normal adhesive properties of the two tissues difference do not appear to be due to cadherin levels, but to a still unknown mechanism, probably involving differences in the actin cytoskeleton[31–33]. Nevertheless, varying cadherin levels remain a simple and effective experimental method to manipulate cell–cell adhesion[6, 13, 27, 28, 34]. We increased cadherin levels by expressing C-cadherin-GFP, or depleted

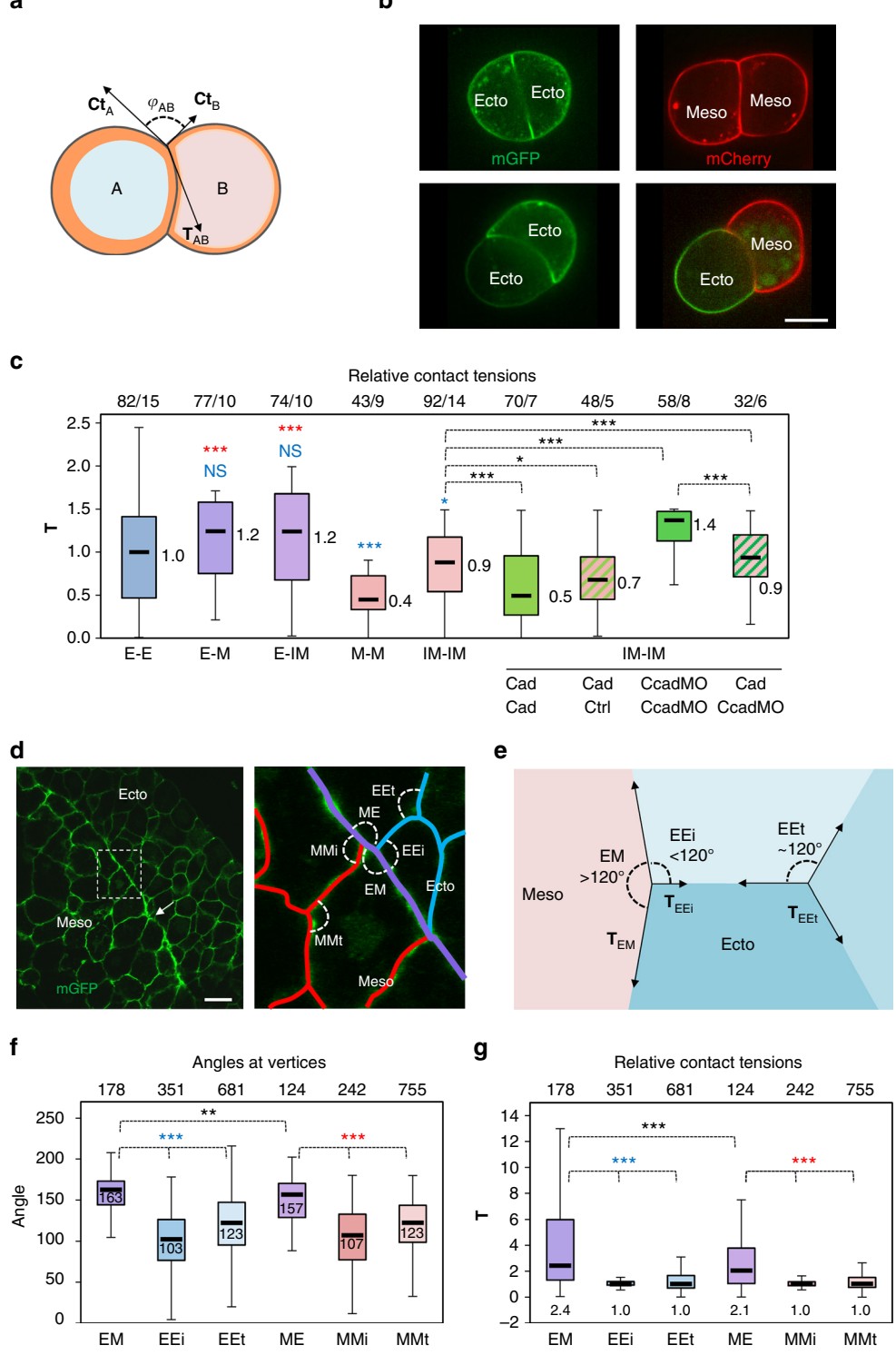

endogenous C-cadherin using an antisense morpholino (MO) (Supplementary Fig. 2a). We chose levels of overexpression/ depletion that led to changes in cell adhesion comparable to those found for endogenous tissues (Supplementary Fig. 2c). Depletion of myosin II heavy chains A and B (MHC) in the ectoderm reduced the elastic modulus to values similar to endogenous mesoderm and IM, whereas expression of constitutively active RhoA (caRho) in IM cells matched the stiffness of ectoderm cells (Supplementary Fig. 2d). Note that cadherin adhesion and actomyosin are interdependent, and altering one of the two systems is bound to affect the other. Despite this complexity, estimates of contact tension allowed us to evaluate the global effect of each manipulation (see below).

**Estimates of tension at homotypic and heterotypic contacts.** Cell and tissue geometry can be used to infer information about their underlying forces (e.g. refs. [7, 12, 13, 23, 24, 28]). Cell doublets provide a simple configuration (Fig. 3a), where the angles at vertices reflect the balance between the contact tension ($T$) and the cortical tension at the free cell surface ($Ct$)[13]. $Ct$ is proportional to the elastic modulus, which was determined by AFM (Fig. 2e). We had to consider heterotypic doublets composed of ectoderm and mesoderm cells with different $Ct$s, geometrically reflected by the curved contact interface (Fig. 3a, b; Supplementary Fig. 3b). Curved interfaces were also frequently found in homotypic doublets (Fig. 3b; Supplementary Fig. 3g), which was consistent with the high variability of the measured elastic modulus (Fig. 2e).

$T$ estimates for the different types of doublets are presented in Fig. 3c. Similar estimates were obtained using a simplified model of symmetrical doublets (Supplementary Fig. 3a, c–e)[6]. $T$ at homotypic contacts appeared higher for ectoderm than mesoderm and IM, confirming previous estimates based on explant surface tension[12, 28]. However, $T$ at heterotypic contacts was higher than both homophilic $T$s. This result is in agreement with the strong ephrin-Eph-dependent activation of myosin observed at the ectoderm–mesoderm boundary[21]. The distribution of heterotypic $T$ showed a secondary peak corresponding to very high tension (arrows in Supplementary Fig. 3f), consistent with the observed oscillations between phases of repulsion and re-adhesion[20, 21].

A second independent estimate of $T$ was made based on angles at vertices within the embryo (Fig. 3d). In an approximate 2D projection of a homogenous tissue, angles should approach 120° (Fig. 3e). Configurations departing from this scheme are indicative of tension imbalance. At a perfectly smooth boundary interface, angles would be close to 180°, reflecting high relative $T$ along the interface. We observed indeed a clear asymmetry along

the boundary, with larger angles formed by heterotypic contacts (EM and ME, Fig. 3f). On the basis of these angles, we calculated that heterotypic $T$ ($T_{EM}$ and $T_{ME}$) was about twice higher than $T$ at homotypic contacts (Fig. 3g). The boundary was not completely smooth (arrow in Fig. 3d), indicative of low local contact tension corresponding to transient phases of reattachment[20]. This was reflected in the bimodal distribution of angles for heterotypic contacts (Supplementary Fig. 3i), with a major peak around 180°, indicative of high interfacial tension, and a minor peak around 120°, indicative of lower tension. We also noticed a difference in the heterotypic $T$ calculated for the two types of vertices occurring along the boundary: $T_{EM}$ was slightly larger than $T_{ME}$, which indicated that mesoderm is less coherent than ectoderm (calculated ratio $T_{EE}/T_{MM}$ of ~1.34, Supplementary Methods), confirming results from cell doublets (Fig. 3c) and previous estimates[12]. Another feature of the boundary was the homogeneity of $T$ at homotypic contacts adjacent to its interface (EEi and MMi) compared to $T$ inside the tissues (EEt and MMt). This observation suggests that the high tension exerted at heterotypic contacts may impose physical constraints on the boundary cells, whereas the tension of individual contacts within a tissue may be freer to fluctuate.

We also used the geometry of cell doublets to estimate the effect of altering cadherin/myosin/Rho on contact tension (Fig. 3c, right part of the graph, and Supplementary Fig. 3d, h). As predicted, tension varied inversely to cadherin levels, whereas it was lowered by myosin depletion and raised by caRho expression. In particular, the tension of myosin-depleted ectoderm cells matched the tension of mesoderm cells, whereas conversely caRho increased mesoderm tension to the levels found for ectoderm, confirming that these manipulations mimicked physiological conditions.

DAH and DITH predicted that heterotypic contacts should have an intermediate tension (Fig. 1c), although this tension had not yet been determine experimentally. Here heterotypic doublets could provide an estimate of this parameter (Fig. 3c; Supplementary Fig. 3h). Starting with IM cells, we considered heterotypic doublets made of one cadherin-overexpressing cell and one normal cell (Fig. 3c, Cad-ctrl) or a cadherin-depleted cell (Cad-CcadMO). In both cases, the calculated $T$ was close to the average of the two homotypic tensions. Similarly, $T$ between normal and caRho-expressing cells fell roughly half way between the tensions of the homotypic contacts (Supplementary Fig. 3h). For ectoderm cells, $T$ between normal and cadherin-depleted cells were also intermediate, but $T$ between normal and myosin-depleted cells remained high (Supplementary Fig. 3h). We do not have yet an explanation for this case, which likely reflects the complexity of the role of myosin in contractility and adhesion.

**Fig. 3** Estimates of relative contact tensions. **a–c** Estimates based on the geometry of cell doublets. **a** Diagram of cell doublet, representing the balance between $Ct_A$, $Ct_B$ and $T_{Ab}$. $\varphi_{AB}$ is the angle formed between $Ct_A$ and $Ct_B$. The orange layer symbolises the actomyosin cortex. A curved cell–cell interface reflects unequal $Ct_A$ and $Ct_B$ tensions. **b** Examples of homotypic and heterotypic doublets, expressing membrane-targeted GFP or Cherry. Scale bar, 15 μm. **c** Relative $T$ for different types of doublets, calculated based on measurements of angles at vertices and on the $Ct$ values obtained by AFM (Fig. 2e) (detailed calculation in Supplementary Methods). E ectoderm, M mesoderm, IM induced mesoderm. Also shown combinations of IM doublets made of control (ctrl), cadherin-overexpressing (Cad) or cadherin-depleted cells (CcadMO). Box plot as above. The numbers on top of the graph are number of measured angles/number of experiments. Individual comparisons were done using one-sided Student's t-test. *P < 0.05, **P < 0.01, ***P < 0.001, NS not significant. A colour code was used to indicate comparison to control mesoderm (red) or to ectoderm (blue). **d–g** Estimates based on the geometry of cells within tissues. **d** Section of dorsal ectoderm and mesoderm expressing mGFP. Right panel: Enlargement of the boundary area, with homophilic contacts highlighted in blue and red, and the heterophilic boundary interface in purple. Angles formed at vertices between different types are indicated: EM and ME, angles between heterotypic contacts; EEi and MMi, angles between heterotypic and homotypic contacts at the tissue interface; EEt and MMt, angles between homotypic contacts inside each tissue. Scale bar, 30 μm. **e** Representation of vector forces at vertices. Angles at a vertex within an ideal homogenous tissue should be around 120°. Tissue boundaries tend to be straighter, thus, EM and ME angles are larger than EEi and MMi angles. Such asymmetry is indicative of higher contact tension. **f, g** Angle measurements and calculated relative $T$ (Supplementary Methods). $T$ at heterotypic contacts is about twice as high as at homotypic contacts. Measurements from 12 embryos. Plots and statistics with colour code as above

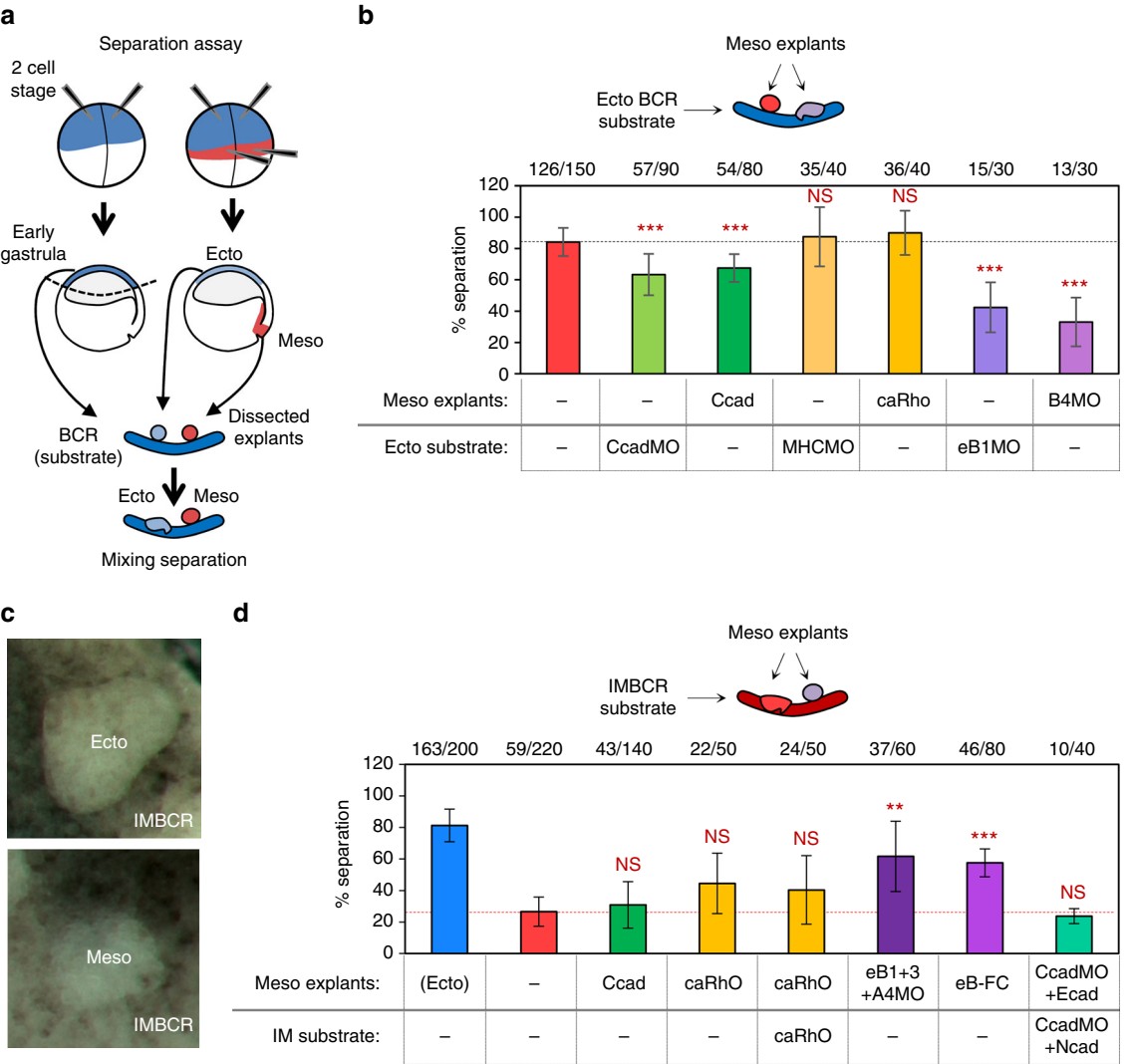

**Fig. 4** Impact of adhesive and contractile difference on ectoderm–mesoderm separation. **a** Scheme of the separation assay. Embryos injected at the 2 cell-stage were dissected at the early gastrula stage. Explants were laid on a BCR, and the number of explants remaining separated were scored. **b** Maintenance of ectoderm–mesoderm separation. Cell–cell adhesion was levelled by C-cadherin depletion in the ectoderm (morpholino injection, CcadMO), or overexpression in the mesoderm (mRNA injection, Ccad). Differences in contractility were levelled by depletion of MHC2A and B in the ectoderm (MHCMO) or by expression of constitutively active RhoA (mRNA injection, caRho) in the mesoderm. Ephrin-Eph signalling was inhibited by depletion of ephrinB1 (eB1MO) or EphA4 (A4MO), respectively, in the ectoderm or the mesoderm. **c** Example of the assay using a blastocoel roof induced to mesoderm as substrate (IMBCR). Ectoderm explants remain separated, whereas mesoderm explants sink into the IMBCR. **d** Induction of separation between mesoderm explants and IMBCR. Differences in cell–cell adhesion were imposed by C-cadherin overexpression (Ccad) and differences in contractility by expression of caRho. Ephrin-Eph signalling was stimulated by expression of ephrinB1 and B3 and simultaneous depletion of EphA4. Alternatively, signalling was activated at the surface of the mesoderm explants using preclustered ephrinB3-Fc soluble fragments (eB-Fc). Differences in cadherin expression were created by C-cadherin depletion (CcadMO) and its replacement with E-cadherin on one side and with N-cadherin on the other side. Ectoderm explants were used as a positive control (*blue column*). In an additional control, myosin activity was increased in both explant and substrate (caRho). The *numbers* on top corresponds to the number of explants that remained separated/total explants (10 explants per independent replicate). Graphs show mean values, error bars s.d. Individual comparisons to control mesoderm (*red asterisks*) were done using one-sided Student's *t*-test

**Impact of adhesive/contractile difference on separation.** Separation was tested using an assay where the boundary was reconstituted by placing explants on a blastocoel roof (BCR) (Fig. 4a)[35]. We have previously shown that separation was decreased by about 50% by ephrin or Eph depletion in one of the two tissues, and almost completely blocked by inhibiting Eph signalling on both sides of the boundary[20, 21]. Figure 4b shows two examples of partial inhibition caused by depletion of ephrinB1 in the ectoderm or EphB4 in the mesoderm.

If the differences in cortical tension observed between ectoderm and mesoderm had a role in the separation of these two tissues, levelling these differences should impair the process.

However, neither myosin depletion in the ectoderm nor caRho expression in the mesoderm had any detectable effect on separation (Fig. 4b).

Dampening the adhesive differences between the two tissues by depleting C-cadherin in the ectoderm or overexpressing it in the mesoderm led to a modest decrease in separation (Fig. 4b). Note that these manipulations of cadherin increased the difference in the relative **T** of these two tissues (Fig. 3c; Supplementary Fig. 3h), a situation which, according to DITH, should have sharpened separation. We wondered whether the weak inhibition resulting from these cadherin manipulations represented a parallel contribution of DAH to reinforce ephrin-dependent

HIT. We thus combined cadherin depletion/overexpression with ephrin/Eph loss-of-function. However, we did not observe any further inhibition of separation compared to ephrin/Eph depletions alone (Supplementary Fig. 4a).

We also examined the effect of levelling adhesion or contractility on the ectoderm–mesoderm boundary in whole embryos (Supplementary Fig. 5). Again, the boundary largely insensitive to these manipulations, whereas it was strongly

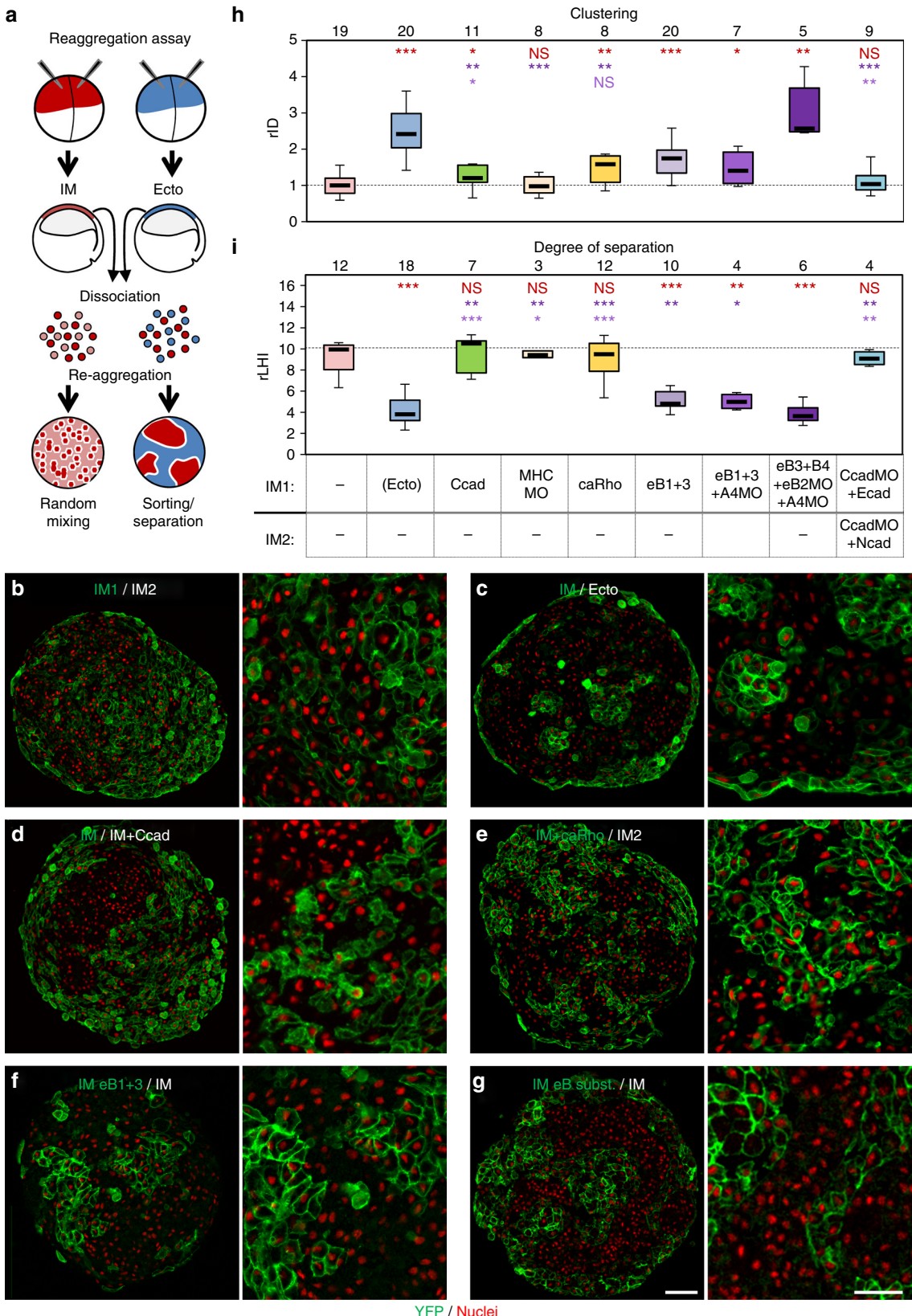

YFP / Nuclei

perturbed by ephrin/Eph depletion[20, 21]. In summary, these data failed to reveal a requirement for adhesive or contractile differences at the ectoderm–mesoderm boundary.

**Requirements for induction of separation**. We next asked whether separation could be ectopically induced by re-creating the differences in adhesion or contractility existing between ectoderm and mesoderm, or by stimulating ephrin-mediated repulsion at their interface. Figure 4c and d presents the results for separation of two mesoderm tissues, which were assayed by combining mesoderm explants with a mesoderm-induced BCR (IMBCR). Cadherin overexpression was not able to stimulate separation. caRho caused only a partial increase in separation. Importantly, the same effect was observed when caRho was expressed in both the explants and the IMBCR, demonstrating that the effect was not due to a tensile difference between the two tissues. To simulate ephrin-Eph signalling occurring across the boundary[21], we ectopically expressed ephrinB1 and B3, and simultaneously depleted EphA4. We verified that tension was increased at heterotypic contacts with control IM but not at homotypic contacts (Supplementary Fig. 3h). We had thus succeeded at reconstituting a genuine situation of HIT. Consistently, these conditions led to robust separation (Fig. 4d). As an alternative approach, we activated Eph at the surface of the explants using pre-clustered ephrinB-Fc fragments[20, 21]. This treatment, which increased cortical stiffness to levels found for ectoderm or caRho-expressing IM cells (Supplementary Fig. 2d), induced robust separation (Fig. 4d).

As C-cadherin is the only major cadherin expressed at this stage, SAH cannot be involved in normal separation of these tissues. However, we could create SAH conditions by replacing endogenous C-cadherin with E- in the mesoderm and with N-cadherin in the IMBCR (Supplementary Fig. 2b). Yet these substitutions did not cause separation (Fig. 4d). In conclusion, induction of ephrin-Eph-mediated repulsion appeared to be the only condition that could efficiently induce separation of mesoderm tissues. Note that the partial separation obtained with caRho could be explained by the fact that ephrin-Eph signalling operates via Rho to locally increase contractility[20].

Experiments with ectoderm tissues led essentially to the same conclusion (Supplementary Fig. 4b): the most efficient separation was again induced by ectopic ephrin-Eph activation. Ectoderm appeared to be more sensitive than mesoderm to manipulations of adhesion and tension, which all caused some separation, but the effect was in all cases stronger in control experiments where both explants and BCR were manipulated, demonstrating that the effect was not due to DAH/DITH.

**Requirements for cell sorting from mixed aggregates**. To test sorting from two mixed populations, we used the classical re-aggregation assay (Fig. 5a)[2]. We measured two parameters, a relative index of dispersion (rID, Fig. 5h and Supplementary Fig. 6a), which provides a quantification of cell clustering, and the relative length of the total heterotypic interface formed between the two cell populations (rLHI, Fig. 5i and Supplementary Fig. 6b), which also takes into account the smoothness of the heterotypic interface[36]. IM and ectoderm cells efficiently sorted from mixed aggregates to form well-segregated groups (Fig. 5c, h, i). Two populations of untreated IM cells remained intermixed and formed a large heterotypic interface (Fig. 5b, h, i). When one of the IM populations was manipulated to increase adhesion (cadherin overexpression), decrease contractility (MHC depletion) or increase contractility (caRho), we observed clustering (Fig. 5d–h). However, the clusters remained extremely irregular, which was reflected by high rLHI (Fig. 5i). Cadherin substitution had no effect (Fig. 5h, i). On the contrary, ephrin expression (with or without EphA4 depletion), not only led to clustering, but also to shortening of rLHI (Fig. 5f, h, i). As sorting under this condition was not as complete as between ectoderm and IM, we set to reconstitute the two antiparallel systems active at the endogenous boundary[21]: we depleted IM cells of ephrinB2 and EphA4, which we replaced with ectodermal ephrinB3 and EphB4 (Fig. 5g, h, i). We now obtained a full separation from control IM cells, with interfaces as smooth as those observed with ectoderm. These results showed that ephrin-Eph-mediated repulsion was the only condition among those tested in this study that could successfully produce tissue segregation. Furthermore, reconstitution of the two antiparallel ephrin-Eph systems was sufficient to fully reproduce the natural sorting of ectoderm and mesoderm.

We complemented these results with an additional assay, which examined the dispersion of a clone of manipulated cells within an IMBCR (Supplementary Fig. 7). Again, stimulation of ephrin-Eph signalling was most efficient at preventing dispersion. caRho expression also decreased dispersion, but the effect was even stronger when caRho was expressed in the whole BCR, indicating that this effect was not driven by DITH, but from a general impairment of dispersion.

**Computer simulation confirms the high efficiency of HIT**. Cell sorting had been effectively modelled based on the principle of contact tension using the cellular Potts model[37], see Supplementary Methods. In this model, cells are connected domains of pixels on a lattice that evolves according to a set of probabilistic rules. Cell-medium and cell–cell contacts are given 'energies', which represent **Ct** and **T**. We used this model to compare sorting in

**Fig. 5** Requirements for cell sorting from mixed aggregates. **a** Diagram of the assay: mixed aggregates composed of unlabelled cells and mGFP-expressing cells were fixed and cryosectioned. GFP-expressing cells were detected with an anti-GFP antibody (*green*) and nuclei were counterstained with Hoechst (*red*). **b–g** Representative examples, with detailed views in *right panels*. **b** Aggregates of unlabelled and labelled IM cells, which distributed randomly. **c** Positive control: GFP-labelled IM cells sorted efficiently from unlabelled ectoderm, forming sharply delimited groups. **d–f** Control IM cells were mixed with IM cells manipulated as follows: **d** C-cadherin overexpression; **e** expression of constitutively active RhoA; **f** EphrinB1+,3 expression; **g** substitution of mesodermal ephrinB2 and EphA4 (eB2MO and A4MO) with ectodermal ephrinB3 (eB3) and EphB4 (B4). Scale bars, main panel 200 μm, enlargement 100 μm. **h** Quantification of cell clustering, using a relative index of dispersion (*rID*). Clustering is defined as departure from a random distribution (index of 1). Box plot as above. The *number* of aggregates is indicated on top. One-sided Mann–Whitney test was used for statistical comparison. Colour coded *asterisks* as follows: *Red*, comparison to control IM; *dark* and *pale purple*, comparison to ephrinB1 + 3 expression and to ephrin/Eph substitution (eB3 + B4 + eB2MO + A4MO). Except for myosin depletion (MHCMO) and cadherin replacement (CcadMO + Ecad/Ncad), all manipulations led to some significant degree of clustering compared to control non-manipulated IM. Ephrins/Eph substitution led to the highest degree of clustering. **i** Quantification of the relative length of heterotypic interface (*rLHI*), defined as the ratio between the total length of heterotypic contacts and the theoretical minimal interface, calculated as the perimeter of a *circle* encompassing the total area of the labelled population (Supplementary Fig. 6). Conditions of ephrin ectopic expression alone or in combination with EphA4 depletion led to a significant shortening of rLHI. Separation was further improved by substitution of mesodermal with ectodermal ephrin/Eph. None of the other conditions significantly departed from the negative control aggregates of non-manipulated IM. Plot, statistical test and colour code as in **h**

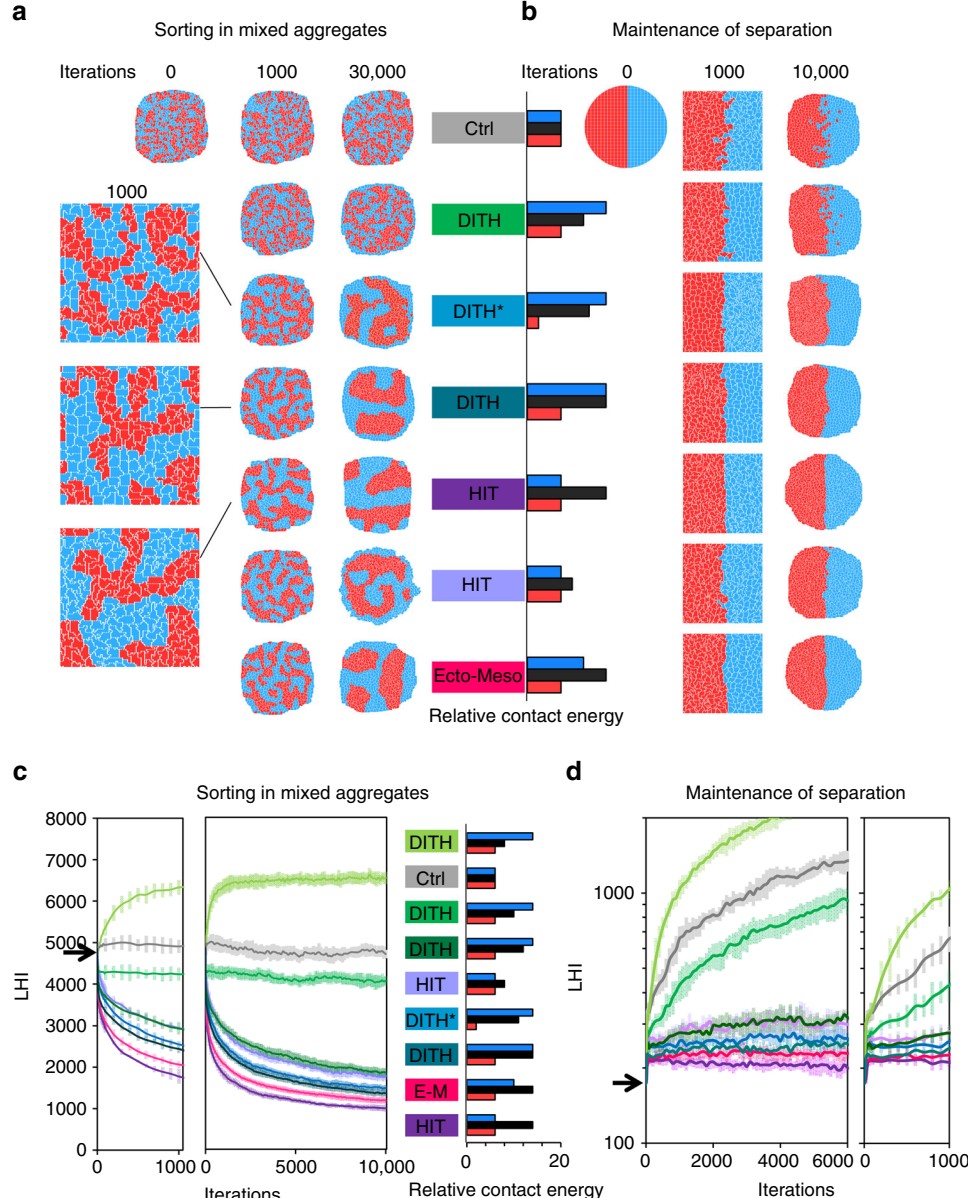

**Fig. 6** Model simulations of cell sorting and tissue separation. Simulations of sorting from a mixed aggregate (**a**, **c**) and of maintenance of a boundary interface (**b**, **d**). **a**, **b** Representative snapshots at the indicated number of iterations. Rectangular images represent enlargements of a portion of the aggregates. Initial matrices are shown at 0 iteration. **c**, **d** Evolution of LHI used as an index for cell sorting and for maintenance of separation. *Small panels* show details of the first 1000 iterations. The *curves* represent the average of 15, respectively, 8, independent simulations. Error bars: s.d. In the case of mixed aggregates, each simulation started from a different initial matrix of randomly distributed cells. The same sets of relative contact energies were used for both types of situations. DITH scenarios differ in terms of the relative heterotypic interfacial energy, set at 25, 50, 75 and 100% of the difference between the homotypic tension energies of the two cell populations. DITH* corresponds to the previously published conditions[37]; HIT conditions were set with identical homotypic energies in both cell types, and a higher interfacial tension; ectoderm–mesoderm (E–M) energies were based on estimated relative tensions in the *Xenopus* system. Cell to medium values were set as follows: **a**, **b** 25 for both cell types, all conditions. In graphs **c** and **d**, unequal values 18/9 for all conditions except for the two HIT conditions, where they were set equal, respectively, 9/9 and 18/18. Varying cell to medium values had little impact on sorting and separation (see systematic comparisons in Supplementary Fig. 8b–g)

HIT or DAH/DITH situations (Fig. 6). Relative contact energies were based on our experimental estimates of relative **T** (*red/black/blue columns* in Fig. 6). Varying cell-medium energies had an effect on the general pattern of the aggregates, but little impact on the efficiency of cell sorting (Supplementary Fig. 8b–d).

To simulate ectoderm–mesoderm segregation, the contact energy was set lowest for mesoderm, intermediate for ectoderm, and highest for heterotypic contacts. These settings led to very efficient sorting (Fig. 6a, c): the length of the heterotypic interface

decreased faster and more completely than under the DAH/DITH conditions used in the original simulations[37], (marked as DITH* in Fig. 6). Sorting based on HIT appeared surprisingly robust: any setting where contact energy was highest at heterotypic contacts led to sorting, irrespective of the relative values of the two homotypic energy contacts (Fig. 6a, c; Supplementary Fig. 8a). Even a slight heterotypic to homotypic difference (energies 6-8-6, Fig. 6a, c and 12-14-12, Supplementary Fig. 8a) performed remarkably well.

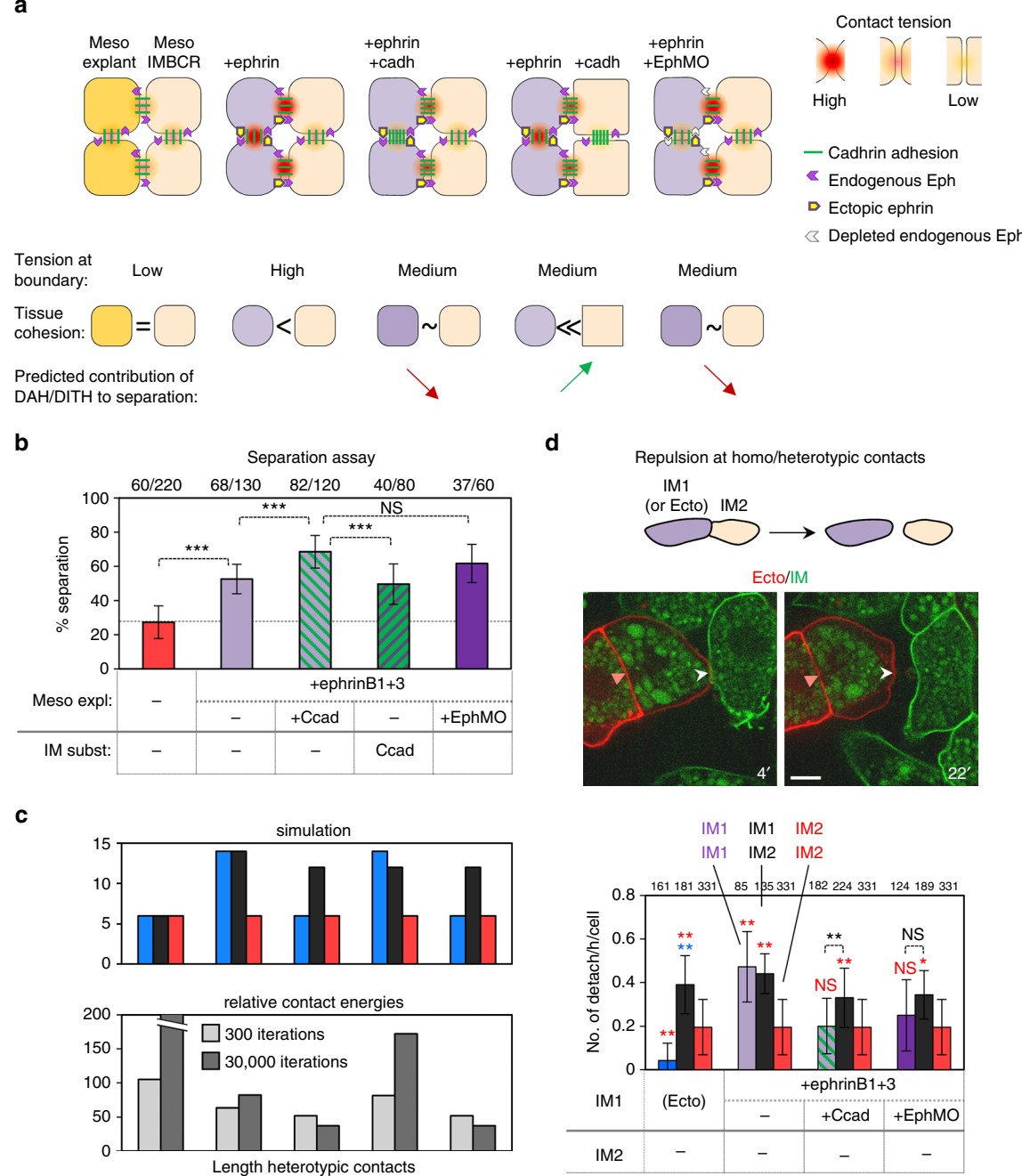

**Fig. 7** Combined effects of tissue cohesion and ephrin-Eph signalling on separation. **a** Graphic representation of the various conditions tested in this experiment. The *upper* diagrams represent the various cell contacts with the tension resulting from the antagonistic action of cadherins and ephrin-Eph receptor pairs. The strength of the contact tension is also reflected in the shape of the cells (*rounder outlines* correspond to higher tension). The *lower part* of the panel summarises the impact of each manipulation on heterotypic interfacial tension and on tissue cohesion (inversely related to homotypic contact tension), and the predicted effect on separation assuming a contribution of DAH/DITH. Normal mesoderm–mesoderm contacts have low contact tension. Ephrin ectopic expression stimulates repulsion both at the explant interface and within the explant, resulting in higher tension at both homotypic and heterotypic contacts. Cadherin overexpression in the explant counteracts ephrin-induced repulsion within the explant, and to a lesser degree at the interface. Cadherin expression on the other side of the boundary decreases tension in the IMBCR, thus enhancing the difference in homotypic tension of the two tissues. EphA4 depletion (EphMO) in the explant reduces ephrin-induced repulsion within the explant (and to a lesser extent at the interface). **b** Result of the separation assay (graph and statistics as in Fig. 4). Ephrin-induced separation was stimulated by cadherin co-expression or Eph depletion, inconsistent with a contribution of DITH. **c** Simulation of maintenance of separation for each experimental condition. Energy values were based on the results shown in **d**. The results of the simulation were in agreement with the experimental data. **d** Cell–cell repulsion quantified as the hourly rate of cell detachment. *Top*: Diagram of the experiment. *Middle*: Two frames of a time-lapse confocal microscopy movie. Ectoderm and IM cells expressed, respectively, membrane Cherry and membrane GFP. The *pink arrowheads* point to a stable homotypic contact, the *white arrowheads* to a heterotypic contact that detached. Scale bar, 10 μm. *Bottom*: Quantification of detachments at homotypic (*blue/purple* and *red columns*) and heterotypic contacts (*black columns*). Statistical comparisons to ectoderm or mesoderm controls are colour coded as above

On the contrary, only a limited range of settings simulating DAH/DITH were able to drive sorting. Importantly, the outcome did not depend on the difference between the homotypic contact energies of the two tissues, but exclusively on the relative strength of the heterotypic energy (Fig. 6; Supplementary Fig. 8a): detectable sorting only occurred when this heterotypic energy was set closer to the highest homotypic energy, as in the original DITH* condition. When the value was set midway, as to match our **T** estimates for IM cells (Fig. 3c), sorting was poor, and when set closer to the lowest homotypic energy, cells dispersed even more than in negative controls. In other words, these simulations predicted that sorting based on DITH/DAH could only occur whether tension at heterotypic contacts was dominated by the most tensile (or less adherent) cell type.

We also simulated maintenance of the boundary between two fully segregated populations (Fig. 6b, d; Supplementary Fig. 8). The results were essentially identical to the simulations of sorting in aggregates: the HIT settings that yielded optimal sorting were also the only conditions that were able to maintain the boundary interface. The boundary was also maintained, albeit less efficiently, by DITH conditions with high heterotypic energy, but was rapidly blurred and lost in other DITH settings.

Although the number of iterations is not linearly related to physical time, it provided a 'timeline' that could be compared to the experimental data. The simulations of ectoderm–mesoderm sorting mimicked quite faithfully the characteristics of cell sorting as observed in real tissues. The global distribution obtained after overnight incubation (Fig. 5) resembled simulations after >10,000 iterations (Fig. 6a), whereas shorter simulations (1000 iterations) reproduced early phases of sorting, including for the appearance of smooth interfaces (Fig. 6a and *arrowheads* in Supplementary Fig. 6c). *Xenopus* gastrulation is completed in a few hours, and in this context the first few hundreds of iterations are most relevant. Interestingly, this is the phase of the simulation that showed the largest differences in sorting efficiencies between the various conditions (Fig. 6c, *left panel*). Note also that while extensive cell mixing in negative control of boundary maintenance only occurred at late steps of simulation, the irregular interface at 1000 iterations accurately mimics the morphology of the tissue interface observed when separation is experimentally perturbed (Supplementary Fig. 5b, see also e.g. refs [35, 38]).

In summary, these simulations fully supported the high efficiency and robustness of HIT-based separation and indicated that a DITH-based mechanism could operate exclusively under very restricted conditions.

**DITH viewed as a suboptimal case of HIT**. Our simulations indicated that the strength of the heterotypic tension was absolutely crucial for sorting and separation, both in HIT and DITH scenarios. By systematically varying the relative contact energies (Supplementary Fig. 8a), we realised that HIT and DITH could be seen as a continuum, where the efficiency of sorting (and of maintenance of separation) was dictated by the differences between the heterotypic tension and each of the homotypic tensions: sorting/separation was most efficient when the two differences were positive, i.e. when the heterotypic tension was higher than the tensions in the two tissues. In the simplest HIT case, where both tissues had the same homotypic tension, the sorting/separation efficiency directly depended on the relative strength of the heterotypic tension. The system also accommodated differences between the tissue tensions, as long as they both remained lower than the heterotypic tension. Sorting/separation was still observed when one of the tissue tension reached and even surpassed the heterotypic tension, but with a decreased efficiency. We hypothesised that the situations that

were previously interpreted as DAH/DITH-based cell sorting represented in fact limit cases of HIT, in which a small negative tension difference between the tissue interface and homotypic contacts within one of the tissues could still be compensated by a large positive difference on the other side.

**Tissue cohesion reinforces HIT independently of DAH/DITH**. So far we have only considered the possibility that differences in tissue tension could reinforce separation through a classical DAH/DITH-based mechanism, which would act in parallel with ephrin-Eph-dependent HIT. However, our simulations suggested a diametrically opposite model, where the most favourable condition would be low tension in both tissues that would contrast with high tension at their interface.

To distinguish between these two hypotheses, we set an experiment that tested the impact of tissue cohesion on ephrin-induced separation between mesoderm tissues (Fig. 7a). We chose ectopic ephrinB1+3 expression (Fig. 7b) as a basal condition of separation. The essential characteristic of this condition was that repulsion was induced both at the tissue interface and between the ephrin-expressing cells (Fig. 7a, d), resulting in lower cohesion of the explant (Supplementary Fig. 9a). We had thus created a situation where contact tension was high both in the explant and the boundary (Fig. 7a, c). We then compared the effect of overexpressing cadherin either in the explants (Fig. 7a), which aimed at antagonizing ephrin-induced repulsion (Fig. 7c, d) and restoring tissue cohesion (Supplementary Fig. 9a), or, on the contrary, in the IMBCR, which should enhance the difference between the two tissues (Fig. 7c). The two hypotheses made clear predictions about the outcome of this experiment: if DAH/DITH contributed to separation, cadherin expression in the ephrin-expressing explants should weaken separation, whereas its overexpression on the other side should boost it (Fig. 7a). Our simulations predicted the exact opposite outcome. The experimental result unambiguously supported the second model (Fig. 7b): reinforcing cohesion of the ephrin-expressing explant further increased separation, whereas expression in the IMBCR had no effect. Note that cadherin co-expression had the same effect as EphA4 depletion in decreasing repulsion at homotypic contacts (Fig. 7d) and increasing separation (Fig. 7b). The notion that the major effect of cadherin overexpression was to restore cohesion of ephrin-expressing explants was supported by an additional control, which showed that the same cadherin overexpression did not increase separation induced by soluble ephrin-Fc, which activated repulsion only at the surface of the explant (Supplementary Fig. 9b).

Importantly, enhanced separation upon cadherin overexpression and EphA4 depletion occurred despite the fact that these two manipulations also partially weakened repulsion at heterotypic contacts (Fig. 6c). This somewhat counterintuitive result was remarkably consistent with our simulations, which predicted that a small difference between hetero and homotypic contacts on both sides of the boundary was more efficient to drive separation than a larger unilateral difference (Supplementary Fig. 8a). In conclusion, the strength of cell–cell adhesion within each tissue is important for separation in so far as it participates in setting the appropriate difference in tension with the boundary interface. Differences between tissues, however, do not seem to contribute to separation.

## Discussion

This study defines the requirements for cell sorting and tissue separation. Contrary to a widespread assumption, classical mechanisms based on differences in adhesion or cortical contractility turned out to be surprisingly ineffective at driving

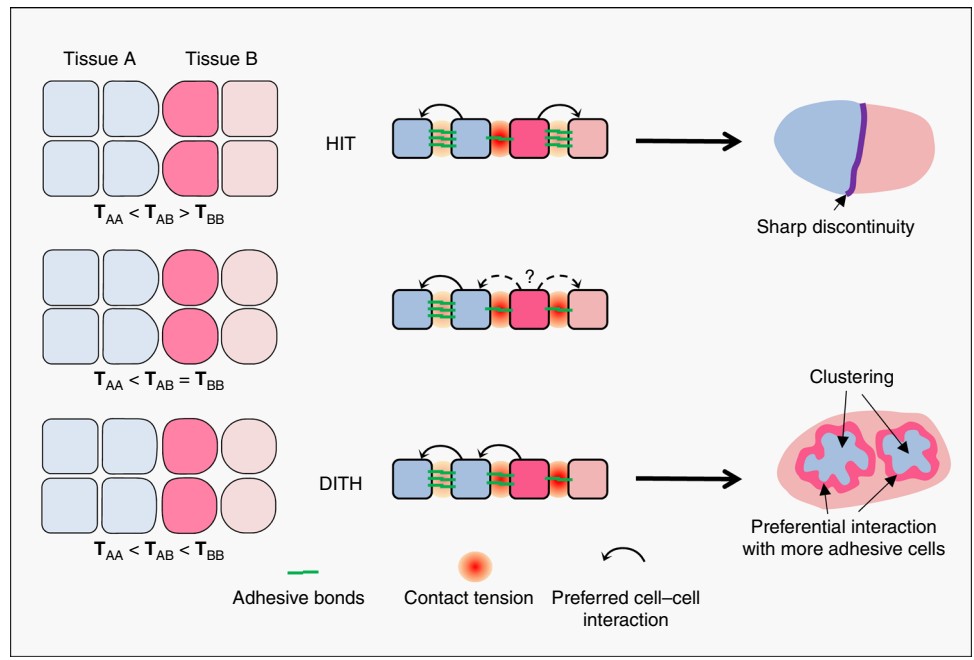

**Fig. 8** Summary diagram comparing HIT and DITH situations. Two cell types A and B are represented in *blue* and *pink*. B-type cells abutting A-type cells are in *darker pink*. In a HIT situation (*top*), where heterotypic contact tension $T_{AB}$ is higher than the two homotypic tensions, cells tend to establish more stable contacts with sibling cells of the same type. At the tissue scale, this drives full segregation of the two cell populations separated by a sharp boundary. In a classical DITH situation (*bottom*), *blue* A-type cells will be able to cluster, but for *dark pink* B-type cells, heterotypic contacts may be more stable than homotypic contacts, thus these cells will tend to preferentially interact with the less tensile/more adhesive A-type cells, a situation that will not lead to clean segregation of the two cell populations. The *middle row* illustrates an intermediate situation, where separation may still be achieved provided a sufficiently strong difference between $T_{AB}$ and $T_{AA}$

cell sorting. The experimental results were fully supported by computational simulations, which further showed that the failure of differences in adhesion/tension to drive tissue separation was not due to some peculiarity of *Xenopus* embryonic cell types, but reflected a more general issue. These simulations predicted that even with large differences in adhesion or tension, sorting may only occur upon fulfilment of a particular requirement: the most tensile (or the lowest adhering) cell should dictate the properties of heterotypic contacts. It is difficult to validate/invalidate this assumption based on theoretical grounds as too little is known about the parameters that control adhesion and tension at cell–cell contacts. Our estimates of contact tension of mesoderm doublets were not consistent with this assumption (Fig. 3c). Wild type vs. myosin-depleted ectoderm was one case where heterotypic tension was particularly high (Supplementary Fig. 3h), thus compatible with contractility-based sorting. Yet, even there, our experiments failed to uncover a clear effect of differential tension (Supplementary Fig. 4b). It will be interesting to identify other cases were a large difference in tissue cohesiveness could lead to separation. We predict that they are unlikely to occur with notable frequency, because they would impose crippling constraints on other morphogenetic processes.

On the contrary, a mechanism based on HIT is effective over a wide range of conditions, regardless of whether the populations have identical or different physical properties. The outcome depends on the difference between the tension at the tissue interface and each of the homotypic tissue tensions. Separation is best achieved when heterotypic tension is higher than both homotypic tensions. Separation can also occur, when one of the homotypic tensions is equal or even slightly higher than the heterotypic tension, provided, however, that this 'negative difference' is compensated by a large difference between the heterotypic and the second homotypic tension. This scenario

corresponds precisely to the conditions previously used to simulate DAH/DITH-based sorting[37]. According to the HIT model, they constitute a particular case of 'asymmetric' HIT.

As depicted in Fig. 8, the logic of sorting and separation can be intuitively grasped by considering the types of contacts established at the tissue interface (*dark pink cell*): in a HIT situation (*top diagrams*), homotypic contacts with a sibling (*light pink cell*) will be in all cases most favourable, leading to efficient sorting and stable maintenance of a sharp boundary. In a classical DAH/DITH scenario, however, a heterotypic contact with a more adhesive (*blue*) cell may be at least as stable (*middle* diagrams) or more favourable than a homotypic contact (*bottom* diagrams). In the best of cases, this will lead to coarse clustering driven by the compaction of the highly adhesive cells and the concomitant partial exclusion of the low adhesive cells. In the worst case, it could even favour dispersion, low adhesive cells trying to maximise heterotypic contacts at the expense of homotypic contacts (see simulations in Figs. 6c, d, *light green* DITH condition).

Previous experimental evidence supporting the models of DAH/DITH failed to take into account important factors: (1) No clear distinction was made between clustering and actual separation. We showed here that DAH/DITH can cause clustering, but only HIT is efficient at producing a sharp boundary, which is required to effectively segregate two tissues. (2) Experiments that interfere with myosin are ambiguous without adequate controls: some phenotypes which could be (mis)interpreted as evidence for DITH may have a different cause, such as impaired cell motility. (3) Although moderate differences in adhesion/tension can incontestably drive cell sorting, at least in vitro, e.g. ref. [34], the time required would likely to be incompatible with the time scale of in vivo morphogenetic processes. (4) Furthermore, in highly dynamic embryonic tissues, the resulting interface would be unstable, unavoidably blurred by further

intercellular migration. HIT appears to be much better suited for rapid and stable segregation. (5) The highly variable cortical tension of individual cells (Supplementary Fig. 3g, see also ref. [6]), likely inherent to tissues undergoing active morphogenetic movements, does not seem suitable to build an efficient mechanism of separation.

Ephrin-Eph-repulsion used as a prototype in this study is not the only pathway that can create HIT. HIT most likely explains the contribution of the vertebrate paraxial protocadherin (PAPC), which acts in parallel to ephrin-Eph to ensure robust separation of mesoderm from ectoderm[39]. PAPC indirectly regulates intercellular adhesion[40], stimulating it at homophilic contacts and decreasing it at heterotypic contacts[39]. Drosophila echinoid protein also induces interfacial tension, probably in an analogous way to PAPC[41, 42]. Theoretically, the SAH could also potentially establish a HIT-based mechanism, provided that cadherins display strong homotypic preference. This is not the case for type I cadherins[43, 44], consistent with the absence of sorting between N-cadherin and E-cadherin-expressing cells, but could occur for other types of cadherins[43].

In conclusion, both experimental data and theoretical considerations clearly point to the absolute need for a local discontinuity in contact tension to build an embryonic boundary. Observations of high tension at insect boundaries[23, 25] support this general principle and future studies will undoubtedly unravel a variety of molecular mechanisms that can create similar conditions.

## Methods

**Frogs**. Husbandry and ethical handling of *Xenopus* were conducted according to guidelines approved by the Canadian Council on Animal Care.

**Buffers**. MBS-H (1×): 88 mM NaCl, 1 mM KCl, 2.4 mM NaHCO₃, 0.82 mM MgSO₄, 0.33 mM Ca(NO₃)₂, 0.33 mM CaCl₂, 10 mM Hepes and 10 µg per ml Streptomycin and Penicillin, pH = 7.4.
Dissociation buffer (88 mM NaCl, 1 mM KCl and 10 mM NaHCO₃, pH = 9.5.

**Antibodies**. Rabbit anti-β-Catenin (H102) (SC7199), rabbit GAPDH (SC25778) from Santa Cruz Biotechnology, and rabbit anti-GFP (A11122), mouse anti-GFP (A11120) from Invitrogen were all used at a 1:1000 dilution. Mouse anti-C-cadherin (5G5, hybridoma supernatant) and rabbit anti-C-cadherin, both generous gift of Dr Barry Gumbiner) were used, respectively, at 1:20 and 1:5000 dilutions[32].

**mRNA and MO**. mRNA were synthesised in vitro from linearised plasmids[19–21]. Embryos were injected animally into both blastomeres at the two-cell stage (to target the BCR), equatorially in the two dorsal blastomeres at the 4-cell stage (to target mesoderm) and in one blastomere at the 32-cell stage for the dispersal assay (below). For 32-cell stage injections, the amount of mRNA or MO was one-fourth of the listed amount. The MO sequences and injected amounts can be found in Supplementary Table 1. A list of mRNA with the injected amounts can be found in Supplementary Table 2.

**RT-PCR**. Extraction, RT and PCR were performed as described in Schohl and Fagotto[45]. The list of primers can be found in Supplementary Table 3.

**Mesoderm induction**. Embryos were injected animally at the two-cell stage with a mixture of mRNA coding for wild-type β-catenin (100 pg) and a constitutively active activin receptor (caActR, 1000 pg).

**Tissue dissociation**. Dissected tissues were dissociated by 5–10 min incubation in dissociation buffer in agarose-coated dishes.

**Live imaging of cell doublets**. Dissociated cells labelled with GAP-Cherry or GAP-YFP were mixed and plated on glass bottom dish containing 1×MBS-H. Images were acquired using a Quorum technologies WaveFX spinning disc confocal mounted on an automated DMI6000B Leica microscope, with a 40x HCX PL APO CS, NA = 1.25 oil objective was used. Overall, 491 and 561 nm diode lasers were used for GFP and Cherry excitation, respectively. Images were collected with EM CCD 512X512 BT camera using the acquisition software Improvision Volocity 3DM.

**Separation and cell sorting assays**. Unless specified otherwise, mRNA and MO were injected at the two-cell stage. Ectoderm explants were dissected from the inner ectoderm layer and mesoderm explants from the lower lip region before the start of involution (stage 10+). They were laid on a dissected BCR or IMBCR, and covered with a coverglass. Each experiment was scored based on the percentage of test explants remaining separate after a 45 min incubation[20, 35]. For in vitro activation of Eph receptors, explants were pre-incubated with preclustered ephrinB2-Fc fragments (40 nM in MBSH) for 15 min at room temperature.

The re-aggregation assay was modified from Townes and Holtfreter[2]. One of the two-cell populations was labelled by injection of NLS-YFP or membrane-YFP mRNA. Tissues were dissociated at stage 10+. Cells were dissociated, mixed and allowed to reaggregate on agarose-coated dishes in 0.5×MBS-H. Aggregates were incubated at 15 °C for 20 h, fixed and processed for cryosectioning and immunostaining[45].

**Measurements of clustering and separation**. Cell clustering was quantified as follows: XY coordinates of nuclei of labelled cells were determined using ImageJ, and clustering was quantified using the index of dispersion (ID) with PaSSaGE v2 software[46]. Briefly, a grid of 100 × 100 pixel quadrats was overlaid on each image of labelled nuclei and the number of nuclei that fell within each quadrat was counted (illustration in Supplementary Fig. 6a). The index of dispersion is calculated from the mean and variance of counts per quadrat. Specifically, the index of dispersion is the variance-to-mean ratio.

$$\text{ID} = \frac{s^2}{\bar{\chi}}.$$

An index of dispersion equal to 1 is expected for random distribution. A value >1 suggests clustering and a value <1 suggests overdispersion. Owing to the round shape of the aggregates within the square shape of the grid, the lowest ID obtained for random controls was around 2. We thus expressed the results as relative ID, setting the value of random controls to 1.

The degree of separation was quantified as relative length of the heterotypic interface (rLHI). LHI was the sum of all heterotypic interfaces, drawn manually and measured using ImageJ. rLHI was obtained by dividing LHI by the perimeter of a theoretical circle corresponding to the total area of the labelled cells (illustration in Supplementary Fig. 6b).

**Dispersal assay**. The dispersal assay involved an initial injection at the two-cell stage to induce the whole animal cap to become IM, followed by a second injection of one blastomere at the 32-cell stage with NLS-YFP mRNA and manipulative factors (mRNAs and/or MO). The distribution of the YFP-labelled cells was analysed at stage 10.5 on dissected animal caps, stained with Hoechst to identify all nuclei. The degree of dispersion was quantified based on the XY coordinates of the YFP-positive nuclei using Delaunay Triangulation. The length of the heterotypic contacts was measured as above.

**Dissociation assay**. Tissues were dissected from stage 10+ embryos. BCRs from three embryos and ectoderm or mesoderm from six embryos were combined to form explants of comparable size. The explants were put in 1×MBSH and allowed to heal into a sphere (1–2 h). The dissociation assay involved a 1 min incubation of the explants in dissociation buffer, immediately followed by physical dissociation by pipetting three times using a 200 µl pipette tip. Images were taken using an MZ16G stereomicroscope (Leica) using a Qimaging camera (MicroPublisher 3.3 RTV), and the number of single cells was determined using ImageJ software. Results varied between embryo batches, consistent with previous data on tissue cohesion[28]. Values were thus standardised for each experiment compared to values obtained for unmanipulated ectoderm.

**Cell detachments assay**. Single dissociated cells expressing membrane-tagged GFP or Cherry were mixed and seeded at low density on fibronectin-coated glass. Cells were imaged (see imaging of cell doublets) at 3.75 min intervals for a total of 75 min. Detachments of homotypic and heterotypic contacts were counted. Results were expressed as a frequency of detachments per cell per hour.

**Atomic force microscopy**. Single dissociated cells were put on a glass bottom dish in 1×MBS-H. They were compressed using a Bioscope Atomic force microscope (AFM) (Veeco) mounted on an inverted optical microscope (Axiovert S100 TV; Zeiss) with a Silicon Nitride cantilever with a nominal spring constant of $K = 0.01$ (MLCT; Veeco) mounted with a 10 µm polystyrene bead (Polysciences). Force-distance (FD) curves were obtained with the Nanoscope software (Digital Instruments). FD curves were plotted as the deflection of the cantilever as it was lowered in the z-axis to make contact with, and compress single cells. The elastic modulus was calculated by fitting the first 200 nm of each FD curve using a Hertzian compression model for a spherical indenter compressing a spherical substrate. Code kindly provided by X.Y. Chau and P. Grütter, McGill University, available at http://spm.physics.mcgill.ca/research-projects/afm-in-fluids/mechanical-properties-of-neurons. Multiple approaches were taken for each cell and the

average elastic modulus for each cell was used as a single data point. Statistical analysis was done using a one-way ANOVA followed by Tukey's HSD post hoc test for individual comparisons.

**Western blots**. All uncropped western blots can be found in Supplementary Fig. 10.

**Statistical analyses**. All data in column graphs are shown as mean values with s.d. Values in the box plots are the median values, the edges of the box correspond to lower and upper quartiles, and the whiskers to maximum and minimal values without outliers. The type of test is indicated in each legend. All Student's *t*-tests were unpaired and based on either equal or unequal variance, which was determined in each case using the *F*-test. The sample sizes were set based on the variability of each assay.

**Mathematical simulations**. Detailed calculation of relative contact tensions, explanations of the simulations using a modified Potts model and the corresponding software can be found in the Supplementary Information.

**Data availability**. The authors declare that all data supporting the findings of this study, including the code for the Potts model, are available within the article and its Supplementary Information files or from the corresponding authors on reasonable request.

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

## Acknowledgements

This work was supported by funds of the CCSRI, CIHR, ANR and Labex Epigenmed to F.F. We warmly thank Dr. Peter Grütter and his team, for access to AFM equipment, training and advice, and in particular XueYing Chua for providing the code for curve analysis.

## Author contributions

L.C., E.Z., L.K. and F.F. performed experiments, and analysed and interpreted data; E.Z. and P.F designed and built the simulation model. L.C. and F.F. wrote the manuscript.

## Additional information

**Competing interests:** The authors declare no competing financial interests.

