## [Peer Review File · Nature Communications]

Reviewers' Comments:

Reviewer #1 (Remarks to the Author)

The manuscript by Canty et al. addresses the contribution of global differences in tissue adhesion or tissue contractility to tissue separation. The authors first characterize the adhesive and contractile properties of *Xenopus* ectoderm and mesoderm. Using in vitro dissociation assays and AFM measurements of single cells the authors find that ectoderm is both stiffer and more adherent than mesoderm. They then estimate relative contact tension between different cell types based on angle measurements between cell doublets and cell geometry within the tissue. Contact tension was found to be approximately twice as high at heterotypic contact compared to homotypic contacts. The authors then aimed at altering cell-cell adhesion and contractility by altering cadherin and Myosin/Rho, respectively and Eph/ephrin signaling and subsequently analyzed the consequences on tissue separation in an explant assay and in a cell sorting assay. They find that changes in Eph/ephrin signaling, but not alterations in cadherin or Myosin/Rho levels have profound influence on tissue separation and cell sorting in the assays. Finally, the authors combine alterations of cadherin levels and Eph/ephrin and find that alteration of cadherin levels can enhance tissue separation induced by Eph/ephrin signaling. The authors conclude that tissue separation is mainly driven by heterotypic contact tension and not by global differences in adhesive or contractile properties.

The authors address an important biological question. The data on the inefficiency of global changes in cadherin, Myosin, or Rho to alter sorting is interesting and underscores the need for a local difference in tension at the tissue interface. However, increases of mechanical tension at tissue boundaries, and their role in the formation of these boundaries, has been previously established both in vertebrates and invertebrates (e.g. Calzolari, 2014, Major and Irvine, 2005, 2006; Landsberg et al., 2009; Monier et al., 2010).

The authors manipulate cadherin or Myosin/Rho levels to test whether global differences in adhesion or contractility, respectively, influence cell sorting or boundary formation. However, cadherins and actomyosin are interdependent making it often difficult to distinguish effects on adhesion versus contractility. Moreover, the authors use in vitro assays to test the resulting effects on cell sorting and boundary formation. It is unclear how these assays relate to the sorting of cells / boundary formation in the embryo. Previous workers e.g. noted that differences in cadherin expression lead to sorting in in vitro assays, but not in the embryo (e.g. Ninomiya et al., 2012). In vivo data on the (inefficiency) of manipulating cadherin levels/Myosin activity on ectoderm/mesoderm separation in embryos would greatly strengthen the manuscript.

Further comments

Introduction: The introduction is missing a clear mentioning that differences in tension at heterotypic cell boundaries have been previously observed at tissue boundaries in vertebrates and insects and that these local differences are required for boundary formation. Moreover, previous data indicating that differences in cell adhesion are insufficient for the separation of ectoderm and mesoderm (e.g. Ogata et al., 2007) should be mentioned.

Fig. 1E'. It is unclear where the boundary between BCR and AxM is.

Fig. 1F. C-cad staining in the tissue would be important to see whether C-cad plasma membrane levels are similar in the different tissue types (as opposed to intracellular vesicles).

Fig. 2A-A'''. What are the red and green markers?

Fig. 2A-A'. What is different between these two "ecto/ecto" scenarios? Cells in A show a straight interface whereas in A' they show a curved interface.

Fig. 2C. What is Cad-MO? Is MO the control morpholino? If yes, why does it significantly increase tension? The authors need to clarify these points.

Fig. 2. Legend should read "(A-C) Estimates of based on..." (not (A-D)).

Fig. 2F. It is unclear whether the values differ for the different types of angles. The authors should provide a statistical analysis.

The legend to Fig. 3 mentions a panel A', which I do not see in the figure.

Legend to Fig. 3 (and 4). "Purple asterisks indicate comparisons with ephrin or Eph MO conditions." Which one? The authors should clarify.

Fig. 3. It is unclear whether the changes of cadherin levels, myosin contractility, ephrin-Eph-signalling and cadherin expression influence relative contact tensions. This is important to be able to interpret the tissue separation experiments. Contact tension should be estimated based on cell geometries as in Fig. 2D-G.

Fig. 3. It is unclear how depletion of C-cad in ectoderm should level cell-cell adhesion between ectoderm and mesoderm, since both germ layers seem to have similar levels of C-cad (see Fig. 1F).

Fig. 3D. Which Eph receptors and ephrins are expressed in meso explants or IM substrate? Is Eph signaling induced throughout explants or substrate or only across border between substrate/explant. The authors should clarify.

Fig.4 G. How is the index of dispersion defined? 1 corresponds to a random distribution. What does, say "10" correspond to? Why is in the control the index not 1, but rather appr. 2?

Fig. 4G'. I am puzzled what is plotted on the y-axis. What I gather from the figure legend is that it should be the ratio of heterotypic contact length and cell perimeter. But then the ratios should be smaller than 1. The authors need to clarify what they have plotted here.

Fig. 4G and G'. Clustering does not correlate with reduced heterotypic contact length. What does this mean?

Fig. 5C. It is unclear what "length heterotypic contacts" on the y-axis mean? What is the unit?

Fig. 5D. It is unclear what is plotted on the y-axis. What does "100" or "1000" mean in the context of "maintenance of separation".

Reviewer #2 (Remarks to the Author)

Signaling by Eph receptors and ephrins to regulate repulsion and adhesion has been shown to underlie cell segregation and border formation in many tissues during vertebrate development. An important aspect of such signaling is that due to complementary expression, strong activation and cell responses occur specifically at the border. This is in contrast to mechanisms in which global differences in adhesion or tension can drive cell segregation. A number of studies from this group and others have suggested that Eph-ephrin signaling is a much stronger mechanism for segregation than, for example, differential adhesion. The current study carries out quantitative measurements and tests the roles of Eph-ephrin mediated repulsion, differential adhesion and differential tension at the mesoderm-ectoderm border. In addition, computer simulations are carried out to test in principle the ability of these mechanisms to drive segregation and prevent intermingling across a border. The major conclusion is that high interface tension generated by Eph-ephrin signaling is a robust mechanism for segregation, whereas global differences in tension can only drive segregation if the cells with strong tension determine the properties of the interface; however, this latter condition is not observed in vivo.

This is an excellent study that significantly advances mechanistic and conceptual understanding of cell segregation. I strongly recommend it for publication in Nature Communications. There are several points that should be addressed to clarify and improve the study.

1. Previous work from this group has shown that bidirectional forward signaling is required at the mesoderm-ectoderm border (EphA4-ephrinB3 in one direction, EphB4/EphB2-ephrinB2 in the other). Bidirectional signaling is a general feature of Eph-ephrin signaling at borders and may be important for preventing each population from migrating into the other. The current work only tests the roles of unidirectional forward signaling by EphA4 and ephrinB3. To generalise the findings, it would be important to test in experiments and simulations whether or not different results are obtained when high interface tension occurs in one or both cell populations.
2. Page 7 / Fig.3 / Suppl Fig.4. In experiments to test a potential role of differential adhesion, knockdown and overexpression experiments are carried out for C-cadherin. It is found that these manipulations disrupt mesoderm-ectoderm separation, but argued that this is not due to a parallel role of differential adhesion because they did not have an additive effect when combined with Eph/ephrin depletion. This is not convincing, as differential adhesion may simply not be sufficient for separation, and its contribution can only be detected in the context of Eph-ephrin repulsion. Experiments and simulations presented later clarify the role of C-cadherin in suppressing homotypic tension and support that differential adhesion does not have a role. I would suggest to tone down the interpretation of Fig.3 / Suppl. Fig. 4.
3. Page 8 / Fig.4. The dispersion index to quantitate cell segregation is not explained and does not seem to be a useful measurement as it is similar in situations where segregation is clearly different. Is there a more robust way to quantitate segregation that complements the measurements of the length of the heterotypic interface?
4. Fig.4. The segregation for IM/ectoderm seems stronger (more compact clusters, smoother border) than when IM is juxtaposed with IM overexpressing ephrinB3 (with/without EphA4 knockdown). Does this reflect that something is missing in the latter manipulation in which EphA4-ephrinB3 is driving segregation, such as lack of bidirectional signaling (point 1 above)?
5. Minor point. Page 3. It is confusing that EphA4 is described as the sole receptor for ephrinB3. It should be clarified that this refers to the finding in previous work that these are expressed in complementary tissues, whereas other receptors for ephrinB3 overlap.

Reviewer #3 (Remarks to the Author)

General comments

This study combines a variety of experimental studies of cell sorting and boundary maintenance using ectoderm and mesoderm from *Xenopus* the early gastrula with computational modelling to provide compelling evidence for the heterotypic interfacial tension (HIT) hypothesis, in contrast to the competing differential adhesion and differential interfacial tension hypotheses (DAH/DITH).

Key results include: differential adhesion and interfacial tension do not play a significant role in ectoderm/mesoderm separation from mixed aggregates, while ephrin-Eph-mediate repulsion can induce separation in a straightforward manner; and that computational simulations predict that DAH/DITH suffice for sorting only under restricted conditions on the relative tensions of homotypic and heterotypic cell contacts, and cannot maintain an embryonic boundary, while HIT leads efficiently to sorting and boundary maintenance. Based on their results, the authors propose a description of tissue segregation that naturally incorporates DAH/DITH as special cases of the more general HIT.

I found this work to be original, interesting and persuasive. The manuscript itself is well presented. The results appear to have broad implications for embryonic boundary formation and maintenance. I did have a small number of specific comments and queries, listed below.

Specific comments

p.5 "This distribution was fully consistent with the observed oscillations between phases of ephrin-mediated repulsion and phases of re-adhesion" - Could the authors comment on whether they would expect to see a similar 'bimodal' distribution of estimated tensions extracted from their simulations?

p.9 "Even the mildest heterotypic to homotypic difference [...] performed remarkably well" and

p.13 "the time required to achieve morphogenetic segregation in vivo" - Could the authors say a bit more about the correlate between the end points in their simulations and their experiments? In particular, does sorting occur more slowly in simulations with a milder heterotypic to homotypic difference, and if so, given the Monte Carlo simulation approach, should this be interpreted as occurring more slowly in 'physical' time?

p.32 "using the index of dispersion" - Please define this measure explicitly.

p.36 "are dictated by a Hamiltonian" - Please state the precise functional form of this Hamiltonian.

p.37 "with a probability based on" - Please state this probability explicitly.

p.37 "cell-to-medium contact energy" - Did the authors explore the robustness of their observed cell sorting or boundary maintenance behaviour to this parameter value?

p.37 "triplicate simulations for each condition tested" - This seems like a very small number, given the speed with which the cellular Potts model can be simulated. To demonstrate the robustness of the simulation results presented, please increase the number of repetitions and display the mean and standard error (or alternative measure of variation) in Figures 5C and 5D.

p.37 "Detailed algorithms are available upon request" - In the interests of open science and reproducibility, I would strongly urge the authors to provide their simulation code as additional supplementary material.

Typos etc.

p.7 "pay any significant role" -> "play any significant role"

p.11 "restore cohesion (thus low homophilic)" -> "to restore cohesion (thus lower homophilic)"

p.19 "at the free surface of between each and" - This wording doesn't quite make sense

p.21 "twice higher than" -> "twice as high as"

p.22 "The number on top correspond" -> "The number on top corresponds"

p.23 "(C,B) Evolution of" -> "(C,D) Evolution of"

p.28 "homotypic tension, set a 1" -> "homotypic tension, set to 1"

p.30 "cell population fully separated" -> "cell populations fully separated"

p.35 "ratios also allowed to calculate" -> "ratios also allowed us to calculate"

p.36 "probabilities from configuration" -> "probabilities from one configuration"

p.36 "Changes that results in" -> "Changes that result in"

Answer to reviewers

We would like to acknowledge the reviewers for their constructive criticisms. We think that the additional experiments that have been asked have significantly strengthened the manuscript. This is particularly the case for the suggestion of reviewer #2 to reconstitute bidirectional ephrin-Eph signaling, which has led to a stunningly robust sorting, and the determination of contact tensions for conditions where adhesion/contractility were manipulated, which, as predicted by reviewer 1, has greatly strengthened the interpretation of our results.

Major changes:

Note that we have split figure 1, since we think that having separate figures for the models and for the characterization of the ectoderm and mesoderm properties was more coherent. Thus the numbers of the subsequent figures have been shifted. We have also added two supplementary figures S5 and S6, again causing a shift for the subsequent numbers.

Effect of alteration of adhesion/tension on contact tension (supplementary Fig.S3D): As asked by reviewer #1, we have completed estimates of contact tension for a variety of conditions. These additional data turned out to be extremely useful to evaluate the impact of our manipulations on cell contacts. The measurements were performed on cell doublets, because we find this method more accurate than measurements in whole tissues. These results are summarized in a new section (page 8), and used to interpret the results of the explant separation and reaggregation assays presented in the subsequent sections.

In vivo phenotypes (supplementary Fig.S5): In response to reviewer #1, we provide data on the effect of changes in cadherin levels, myosin activity and ephrin/Eph depletion on the endogenous boundary in whole embryos. For this purpose, we performed targeted injections to manipulate these parameters only in the ectoderm or in the mesoderm. The embryos were fixed, sectioned and immunostained in order to produce high quality images of the boundary and to determine with precision the position of the progeny of the injected cells, marked by expression membrane GFP. The results confirm that conditions that equalized adhesion or cortical tension of the two tissues had little to no effect on the boundary. I have added some comments about these experiments below in the detailed answer to reviewers.

Reconstitution of bidirectional ephrin-Eph signaling (Reviewer #2). In our comparison of cell sorting based on DAH, DITH or HIT, we had used for the latter a minimal stimulation of ephrin-Eph signaling. Sorting was superior to any of the DAH/DITH conditions, but was clearly not as complete as what observed between endogenous tissues. Reviewer #2 suggested that reconstituting antiparallel bidirectional signaling would improve sorting. This turned out to be the case, and led to complete segregation of two mesoderm populations. Thus bidirectional ephrin-Eph signals are sufficient to fully account for ectoderm-mesoderm segregation.

Reviewers' comments:

Reviewer #1 (Remarks to the Author):

The manuscript by Canty et al. addresses the contribution of global differences in tissue adhesion or tissue contractility to tissue separation. The authors first characterize the adhesive and contractile properties of *Xenopus* ectoderm and mesoderm. Using in vitro dissociation assays and AFM measurements of single cells the authors find that ectoderm is both stiffer and more adherent than mesoderm. They then estimate relative contact tension between different cell types based on angle measurements between cell doublets and cell geometry within the tissue. Contact tension was found to be approximately twice as high at heterotypic contact compared to homotypic contacts. The authors then aimed at altering cell-cell adhesion and contractility by altering cadherin and Myosin/Rho, respectively and Eph/ephrin signaling and subsequently analyzed the consequences on tissue separation in an explant assay and in a cell sorting assay. They find that changes in Eph/ephrin signaling, but not alterations in cadherin or Myosin/Rho levels have profound influence on tissue separation and cell sorting in the assays. Finally, the authors combine alterations of cadherin levels and Eph/ephrin and find that alteration of cadherin levels can enhance tissue separation induced by Eph/ephrin signaling. The authors conclude that tissue separation is mainly driven by heterotypic contact tension and not by global differences in adhesive or contractile properties.

The authors address an important biological question. The data on the inefficiency of global changes in cadherin, Myosin, or Rho to alter sorting is interesting and underscores the need for a local difference in tension at the tissue interface. However, increases of mechanical tension at tissue boundaries, and their role in the formation of these boundaries, has been previously established both in vertebrates and invertebrates (e.g. Calzolari, 2014, Major and Irvine, 2005, 2006; Landsberg et al., 2009; Monier et al., 2010).

The authors manipulate cadherin or Myosin/Rho levels to test whether global differences in adhesion or contractility, respectively, influence cell sorting or boundary formation. However, cadherins and actomyosin are interdependent making it often difficult to distinguish effects on adhesion versus contractility. Moreover, the authors use in vitro assays to test the resulting effects on cell sorting and boundary formation. It is unclear how these assays relate to the sorting of cells / boundary formation in the embryo.

We mention more explicitly the interdependence between adhesion and contractility (end of section on experimental manipulations, top page 7).

Note that throughout the manuscript, we consider DAH and DITH as two variants of a single model, thus a dissection of the relative contributions of adhesion and contractility is not essential for our conclusions.

Note also that our estimates of contact tension have now been used systematically to evaluate the effect of manipulating cadherin levels and myosin activity.

Previous workers e.g. noted that differences in cadherin expression lead to sorting in in vitro assays, but not in the embryo (e.g. Ninomiya et al., 2012). In vivo data on the (inefficiency) of manipulating cadherin levels/Myosin activity on ectoderm/mesoderm separation in embryos would greatly strengthen the manuscript.

We have added, as mentioned above, a new figure (Fig.S5) presenting in vivo phenotypes. We would like to emphasize, however, that the interpretation of these in vivo experiments is complicated by the fact that formation of the endogenous boundary is tributary of proper gastrulation movements (in particular mesoderm involution), which are sensitive to alterations of adhesion and myosin activity. A particularly striking example presented in Fig.S5H' is the frequent failure of caRho-expressing mesoderm cells to involute, causing their accumulation in the blastopore lip.

We thus favor the reconstituted boundary of the explant-based assay, which allows to test specifically for defects in separation independently of other morphogenetic phenotypes. Note also that we systematically use explants taken just before the start of involution (references in supplementary Material and Methods), in order to insure that we always compare the exact same region of the embryo, even in cases where involution would be impaired.

We faced a second issue with the in vivo experiments: Unlike previous experiments by Ninomiya et al, where the entire dorsal region received cadherin morpholinos, we had here to try to target either the ectoderm or the mesoderm. While a rough targeting of these tissues is easy, perfectly unilateral targeting is difficult, due to the variance in distribution of the progeny of the early blastomeres. We have here been quite stringent, keeping for the analysis only those embryos where the distribution of labelled cells was sufficiently clear-cut, thus warrant of effective dampening of adhesive/tensile differences between the two tissues.

Despite these caveats, the results were fully consistent with the explant-based assay.

Further comments

Introduction: The introduction is missing a clear mentioning that differences in tension at heterotypic cell boundaries have been previously observed at tissue boundaries in vertebrates and insects and that these local differences are required for boundary formation. Moreover, previous data indicating that differences in cell adhesion are insufficient for the separation of ectoderm and mesoderm (e.g. Ogata et al., 2007) should be mentioned.

We have completed this section, adding references for both vertebrate and Drosophila.

Fig. 1E'. It is unclear where the boundary between BCR and AxM is. Clarification

We have now added more detailed indications, including a thin line highlighting the boundary. Note that most of the axial mesoderm has not yet involuted but is still in the blastopore lip, thus does not face the BCR at this stage.

Fig. 1F. C-cad staining in the tissue would be important to see whether C-cad plasma membrane levels are similar in the different tissue types (as opposed to intracellular vesicles).

Cadherin/ β -catenin double staining has been added in Fig.S1.

Fig. 2A-A'''. What are the red and green markers? Clarification

We have added labels.

Fig. 2A-A'. What is different between these two "ecto/ecto" scenarios? Cells in A show a straight interface whereas in A' they show a curved interface. Clarification

The two different images for ecto-ecto doublets were commented in the results, but perhaps the mention was too cryptic. We have now clarified. This is an important observation of this study: even within tissues, cortical tension is highly variable (see also AFM data), which is the reason for curved interfaces (also observed in whole embryos).

We have now added a new histogram (Fig.S3C) that shows the distribution of the estimated tension ratios for ecto-ecto and IM-IM homotypic doublets. Note that we have obtained a similar distribution for doublets of endogenous mesoderm, although the total number of doublets is much smaller (not shown).

This variability per se is an additional argument against the robustness of an hypothetical mechanism of tissue separation that would rely on differential tension, which we now mention in the discussion.

Fig. 2C. What is Cad-MO? Is MO the control morpholino? If yes, why does it significantly increase tension? The authors need to clarify these points. Clarification

No, MO corresponded to cadherin MO. To clarify, we have modified the abbreviations throughout the manuscript and modified the text.

Fig. 2. Legend should read "(A-C) Estimates of based on..." (not (A-D)).

Corrected

Fig. 2F. It is unclear whether the values differ for the different types of angles. The authors should provide a statistical analysis.

Statistical comparison is provided.

The legend to Fig. 3 mentions a panel A', which I do not see in the figure.

Corrected

Legend to Fig. 3 (and 4). "Purple asterisks indicate comparisons with ephrin or Eph MO conditions." Which one? The authors should clarify. Clarification

We have carefully revised all figures and legends, in particular to clarify the various statistical comparisons.

Fig. 3. It is unclear whether the changes of cadherin levels, myosin contractility, ephrin-Eph-signalling and cadherin expression influence relative contact tensions. This is important to be able to interpret the tissue separation experiments. Contact tension should be estimated based on cell geometries as in Fig. 2D-G.

As mentioned above, we have now added a whole new set of estimates of tension for the various experimental conditions (Fig.S3D). We have chosen to measure angles for cell doublets, because this approach is more accurate than the measurement of angles in whole tissues.

Fig. 3. It is unclear how depletion of C-cad in ectoderm should level cell-cell adhesion between ectoderm and mesoderm, since both germ layers seem to have similar levels of C-cad (see Fig. 1F). Clarification

We have added an explanation about this approach. It is indeed artificial, but remains to our knowledge the simplest and most efficient way to modulate adhesion. For instance, we have previously shown (Maghzal et al, Dev Cell 2013) that excess cadherin can fully rescue a strong defect in adhesion (complete cell dissociation), even though this phenotype was the indirect result of exacerbated actomyosin contractility. The estimates of contact tension (Fig.3C and S3D) are consistent with the expected effect on cell adhesion.

Fig. 3D. Which Eph receptors and ephrins are expressed in meso explants or IM substrate? Is Eph signaling induced throughout explants or substrate or only across border between substrate/explant. The authors should clarify. Clarification

We agree that a clear summary of ephrin/Eph expression was indeed missing. We have modified the corresponding section in the introduction (page 4). We also mention our previous observations about their weaker activities within the tissues (Rohani et al 2014).

We have also modified the result section in order to clarify our manipulations and their effects. Thanks to the new estimates of cell doublet contact tension asked by the reviewer (Fig.S3D), we have now been able in particular to verify that ectopic ephrin expression coupled to EphA4 depletion in the IM led indeed to the successful reconstitution of a HIT condition by increasing heterotypic contact tension between with non-manipulated IM cells without affecting homotypic contact tension.

Fig.4 G. How is the index of dispersion defined? 1 corresponds to a random distribution. What does, say "10" correspond to? Why is in the control the index not 1, but rather appr. 2?

The calculation of the index of dispersion is now better defined and explained. We agree that the previous display was confusing, since indeed control random aggregates gave a value around 2. This is due to the fact that the aggregates are round, while the grid used by this method is square (we now show an example in supplementary Fig.S6A). Thus confinement to a round aggregates is trivially considered as "clustering". To display the data in a simpler/clearer way, we now expressed them as relative index of dispersion, simply by setting the median value

for random controls to 1 and calculating the ratio for the other conditions. Thus the scale has changed, but the results remain identical.

Fig. 4G'. I am puzzled what is plotted on the y-axis. What I gather from the figure legend is that it should be the ratio of heterotypic contact length and cell perimeter. But then the ratios should be smaller than 1. The authors need to clarify what they have plotted here. Clarification

We have further clarified the principle of this measurement in the text and legends and we have added diagrams in supplemental Fig.S6B.

We measure the total length of the all heterotypic contacts. This length depends on the number of individual clusters, on the convolution of their outlines, and also on the presence of single cells or small groups of cells of the other cell type within the clusters. A short interface reflects good separation. Because of the high variability of configurations, and to compensate for differences in aggregate sizes and proportions of the two cell populations, we calculate the RELATIVE length of the heterotypic interface as follows: We imagine a virtual disc made by all the labelled cells gathered into an ideal single sorted group (minimal interface). We thus calculate a circle containing the total area covered by the labelled cells, and we deduce its theoretical perimeter. The relative heterotypic interface is obtained by dividing the total length of the actual heterotypic interfaces by this perimeter. A ratio of 1 would mean a perfectly sorted group forming a circle.

Note that we now use the abbreviations LHI for length of heterotypic interface and rLHI for relative LHI throughout the manuscript.

Fig. 4G and G'. Clustering does not correlate with reduced heterotypic contact length. What does this mean?

This point is related to the previous one: The two indices have a quite different sensitivity to various types of distribution. Cells may be clustered yet display a large heterotypic interface. The total length of the interface is a measurement that is very sensitive to the smoothness of the interface, as well as to the presence of single or small groups of missorted cells (see illustration in Fig.S6B). We think that it is useful to keep both indices, since they indeed reflect what we observed in experimental aggregates: cadherin or myosin modifications clearly led to clustering (Fig.5D,E). Yet the interfaces remained highly irregular, and numerous cells remained "trapped" inside clusters, while in the ephrin/EphA4 condition the clusters were more compact (Fig.5F).

Obviously the new condition obtained by reconstitution of bidirectional ephrin-Eph signaling led to a much more complete segregation (Fig.5G). However the incomplete segregation of the original unidirectional ephrin-Eph condition remains quite interesting, because it is more comparable to the cadherin/Rho conditions in terms of "clustering", and allows to detect fundamental differences between the two modes of "sorting".

Fig. 5C. It is unclear what "length heterotypic contacts" on the y-axis mean? What is the unit? Clarification

See above

Fig. 5D. It is unclear what is plotted on the y-axis. What does “100” or “1000” mean in the context of “maintenance of separation”. Clarification

These numbers correspond to the number of Montecarlo iterations, which should be viewed as the “timeline” of evolution of the system. Note that this timeline should not be taken as linearly related to the “real” biological time (see answer to rev #3).

Reviewer #2 (Remarks to the Author):

Signaling by Eph receptors and ephrins to regulate repulsion and adhesion has been shown to underlie cell segregation and border formation in many tissues during vertebrate development. An important aspect of such signaling is that due to complementary expression, strong activation and cell responses occur specifically at the border. This is in contrast to mechanisms in which global differences in adhesion or tension can drive cell segregation. A number of studies from this group and others have suggested that Eph-ephrin signaling is a much stronger mechanism for segregation than, for example, differential adhesion. The current study carries out quantitative measurements and tests the roles of Eph-ephrin mediated repulsion, differential adhesion and differential tension at the mesoderm-ectoderm border. In addition, computer simulations are carried out to test in principle the ability of these mechanisms to drive segregation and prevent intermingling across a border. The major conclusion is that high interface tension generated by Eph-ephrin signaling is a robust mechanism for segregation, whereas global differences in tension can only drive segregation if the cells with strong tension determine the properties of the interface; however, this latter condition is not observed in vivo.

This is an excellent study that significantly advances mechanistic and conceptual understanding of cell segregation. I strongly recommend it for publication in Nature Communications. There are several points that should be addressed to clarify and improve the study.

1. Previous work from this group has shown that bidirectional forward signaling is required at the mesoderm-ectoderm border (EphA4-ephrinB3 in one direction, EphB4/EphB2-ephrinB2 in the other). Bidirectional signaling is a general feature of Eph-ephrin signaling at borders and may be important for preventing each population from migrating into the other. The current work only tests the roles of unidirectional forward signaling by EphA4 and ephrinB3. To generalise the findings, it would be important to test in experiments and simulations whether or not different results are obtained when high interface tension occurs in one or both cell populations.

The issue of bidirectional signaling is an excellent point that we have addressed, as explained above. Indeed, bidirectional signaling turned out to be more efficient than unidirectional signaling at driving cell sorting.

We could not address this issue in simulations, because the model is based on contact tension, which is a global force that results from the integration of adhesion, cortical contractility and repulsive reactions, from both sides. Thus increasing

repulsion is equivalent to increase tension. We have tested a wide range of conditions (see Fig.S8), which encompass virtually any possible scenario. For instance, high repulsion within both tissues (but higher at the boundary) may correspond to the condition 12-14-12.

2. Page 7/ Fig.3 / Suppl Fig.4. In experiments to test a potential role of differential adhesion, knockdown and overexpression experiments are carried out for C-cadherin. It is found that these manipulations disrupt mesoderm-ectoderm separation, but argued that this is not due to a parallel role of differential adhesion because they did not have an additive effect when combined with Eph/ephrin depletion. This is not convincing, as differential adhesion may simply not be sufficient for separation, and its contribution can only be detected in the context of Eph-ephrin repulsion. Experiments and simulations presented later clarify the role of C-cadherin in suppressing homotypic tension and support that differential adhesion does not have a role. I would suggest to tone down the interpretation of Fig.3 / Suppl. Fig. 4.

We agree with the reviewer. We have now made sure that these results are presented in a neutral style.

3. Page 8 / Fig.4. The dispersion index to quantitate cell segregation is not explained and does not seem to be a useful measurement as it is similar in situations where segregation is clearly different. Is there a more robust way to quantitate segregation that complements the measurements of the length of the heterotypic interface?

We now explain the index of dispersion in the material and method, and shortly in the legend. We had already tried different parameters for this measurements and considered different methods. As mentioned in the response to reviewer #1, we still consider that it adequately reflects the reality of these aggregates, where cadherin or myosin manipulations create obvious clusters with highly irregular outlines.

4. Fig.4. The segregation for IM/ectoderm seems stronger (more compact clusters, smoother border) than when IM is juxtaposed with IM overexpressing ephrinB3 (with/without EphA4 knockdown). Does this reflect that something is missing in the latter manipulation in which EphA4-ephrinB3 is driving segregation, such as lack of bidirectional signaling (point 1 above)?

See above

5. Minor point. Page 3. It is confusing that EphA4 is described as the sole receptor for ephrinB3. It should be clarified that this refers to the finding in previous work that these are expressed in complementary tissues, whereas other receptors for ephrinB3 overlap.

Yes, corrected. The description of the ephrin-Eph pairs active at this boundary has been expanded in this paragraph.

Reviewer #3 (Remarks to the Author):

General comments

This study combines a variety of experimental studies of cell sorting and boundary maintenance using ectoderm and mesoderm from *Xenopus* the early gastrula with computational modelling to provide compelling evidence for the heterotypic interfacial tension (HIT) hypothesis, in contrast to the competing differential adhesion and differential interfacial tension hypotheses (DAH/DITH).

Key results include: differential adhesion and interfacial tension do not play a significant role in ectoderm/mesoderm separation from mixed aggregates, while ephrin-Eph-mediate repulsion can induce separation in a straightforward manner; and that computational simulations predict that DAH/DITH suffice for sorting only under restricted conditions on the relative tensions of homotypic and heterotypic cell contacts, and cannot maintain an embryonic boundary, while HIT leads efficiently to sorting and boundary maintenance. Based on their results, the authors propose a description of tissue segregation that naturally incorporates DAH/DITH as special cases of the more general HIT.

I found this work to be original, interesting and persuasive. The manuscript itself is well presented. The results appear to have broad implications for embryonic boundary formation and maintenance. I did have a small number of specific comments and queries, listed below.

Specific comments

p.5 "This distribution was fully consistent with the observed oscillations between phases of ephrin-mediated repulsion and phases of re-adhesion" - Could the authors comment on whether they would expect to see a similar 'bimodal' distribution of estimated tensions extracted from their simulations?

Unfortunately we did not yet manage to incorporate temporal oscillations in our Potts model, which would be required to see such bimodal distribution.

p.9 "Even the mildest heterotypic to homotypic difference [...] performed remarkably well" and p.13 "the time required to achieve morphogenetic segregation in vivo" - Could the authors say a bit more about the correlate between the end points in their simulations and their experiments? In particular, does sorting occur more slowly in simulations with a milder heterotypic to homotypic difference, and if so, given the Monte Carlo simulation approach, should this be interpreted as occurring more slowly in 'physical' time?

We now comment about this interesting issue in the revised manuscript. Although the number of iterations cannot be considered as linearly related to "physical time", it does reflect the chronology of events. We agree with the reviewer: a shorter interface after a small number of iterations can be indeed related to faster sorting, and milder heterotypic to homotypic difference are indeed predicted to lead to slower sorting, and, conversely, to faster loss of separation when the simulation starts from a boundary. We have now added in Fig.6C and D two enlargements of

the early parts of the simulations, where these differences are most striking (C' and D').

In order to add a temporal aspect to our experiments, we have included in Fig.S6 images of early reagggregates of mixed ectoderm and mesoderm. The early phase of sorting resemble the configurations obtained after short simulations (a few hundreds to 1000 iterations).

p.32 "using the index of dispersion" - Please define this measure explicitly.

Now defined, see answers to other reviewers.

p.36 "are dictated by a Hamiltonian" - Please state the precise functional form of this Hamiltonian.

p.37 "with a probability based on" - Please state this probability explicitly.

We have now expanded the section on the model. We have kept a simple general explanation of the principles for the non-specialist reader, which also mentions the parameter used, followed by a second detailed section where the detailed algorithms are presented.

p.37 "cell-to-medium contact energy" - Did the authors explore the robustness of their observed cell sorting or boundary maintenance behaviour to this parameter value?

We had already tested varying this parameter, and did not find a strong effect. All the simulations presented in the original manuscript used equal values (25/25).

Yet, stimulated by the reviewer's query, we have decided to present it in some detail.

As reminder, this energy reflects the cortical tension at free cell surfaces (Ct). For the scenario mimicking the endogenous ectoderm and mesoderm, the two cell types have different Cts, thus should have different cell to medium energies. The same should be applicable for other conditions with tissues with different contact tensions (DITH), but not for the HIT conditions, where cell to medium energies should be equal.

We have included in supplemental Fig.S8 a systematic comparison of selected conditions with different cell to medium energies, both equal and unequal (panels B-C), and included a few examples of snapshots (D).

We mention in the manuscript that Cts are close to contact tensions, but that the variability of calculated Ct/T of individual doublets was too high to set precise values to these cell to medium energies. We used in our comparisons values ranging from 8 to 25, and more specifically for unequal values 18/9 and 16/8 (Fig.S8B,C).

Both sorting and maintenance of separation appeared to be largely indifferent to cell to medium values, whether set equal or unequal. Varying these energies had however the expected effect on the global shape of the explants, which became

more irregular when the energy (=tension) was lowered (examples now shown I Fig.S8D).

In the main figure 5, we have replaced the curves of the panels C and D with simulations using the most realistic energies, unequal for DITH and Ecto-Meso (18/9), equal for HIT.

We have decided to keep the original values (25/25) for the snapshot of Fig.5A,B, because we felt that differences in the shape of the aggregates was a visual distraction from the main point, which was the tissue interface.

p.37 "triplicate simulations for each condition tested" - This seems like a very small number, given the speed with which the cellular Potts model can be simulated. To demonstrate the robustness of the simulation results presented, please increase the number of repetitions and display the mean and standard error (or alternative measure of variation) in Figures 5C and 5D.

We now present in Fig.5C,D curves resulting from the average of multiple simulations. Note that in the case of reaggregates (C), each replicate started from an independent random matrix. In D, we have only one initial matrix, but run several independent simulations. We have included error bars.

p.37 "Detailed algorithms are available upon request" - In the interests of open science and reproducibility, I would strongly urge the authors to provide their simulation code as additional supplementary material.

The entire software, which runs in java, cannot be simply pasted in the supplemental word file, as it contains separate files and folders. We will be pleased to include it as additional supplemental material, but would need to discuss about the practical aspects with the editor. If this turns out to be problematic, we would obviously make it available upon request.

Typos etc.

All points have been corrected.

p.7 "pay any significant role" -> "play any significant role"

p.11 "restore cohesion (thus low homophilic)" -> "to restore cohesion (thus lower homophilic)"

p.19 "at the free surface of between each and" - This wording doesn't quite make sense

p.21 "twice higher than" -> "twice as high as"

p.22 "The number on top correspond" -> "The number on top corresponds"

p.23 "(C,B) Evolution of" -> "(C,D) Evolution of"

p.28 "homotypic tension, set a 1" -> "homotypic tension, set to 1"

p.30 "cell population fully separated" -> "cell populations fully separated"

p.35 "ratios also allowed to calculate" -> "ratios also allowed us to calculate"

p.36 "probabilities from configuration" -> "probabilities from one configuration"

Reviewers' Comments:

Reviewer #1:

Remarks to the Author:

The authors have satisfactorily addressed my comments. The additional experiments, in particular the measurements of contact tension under different conditions and the influence of changing Cadherin, Myosin and Eph signaling levels on boundary shape in embryos, have strengthened the manuscript.

Reviewer #2:

Remarks to the Author:

The authors have added new data and made changes to the text to clarify various issues. These changes have improved the manuscript and fully address the points raised in my review. In particular, it is very nice to see that the reconstituted bidirectional signaling gives stronger segregation and sharpening than unidirectional. This is an excellent study that significantly advances understanding of contact tension as a robust mechanism for cell segregation.

Reviewer #3:

Remarks to the Author:

The authors have comprehensively addressed all of my concerns in their revised manuscript. I fully support the publication of this work.